# Controlled Visual Hallucination via Thalamus-Driven Decoupling Network for Domain Adaptation of Black-Box Predictors

**Yuwu Lu**[* †]
School of Artificial Intelligence
South China Normal University
Foshan, Guangdong, China
luyuwu2008@163.com

**Chunzhi Liu**[†]
School of Artificial Intelligence
South China Normal University
Foshan, Guangdong, China
2023024323@m.scnu.edu.cn

## Abstract

Domain Adaptation of Black-box Predictors (DABP) transfers knowledge from a labeled source domain to an unlabeled target domain, without requiring access to either source data or source model. Common practices of DABP leverage reliable samples to suppress negative information about unreliable samples. However, there are still some problems: *i) Excessive attention to reliable sample aggregation leads to premature overfitting; ii) Valuable information in unreliable samples is often overlooked.* To address them, we propose a novel spatial learning approach, called *Controlled Visual Hallucination via Thalamus-driven Decoupling Network* (CVH-TDN). Specifically, CVH-TDN is the first work that introduces the thalamus-driven decoupling network in the visual task, relying on its connection with hallucination to control the direction of sample generation in feature space. CVH-TDN is composed of Hallucination Generation (HG), Hallucination Alignment (HA), and Hallucination Calibration (HC), aiming to explore the spatial relationship information between samples and hallucinations. Extensive experiments confirm that CVH-TDN achieves SOTA performance on four standard benchmarks.

## 1 Introduction

Traditional unsupervised domain adaptation (UDA) adapts models trained on a fully labeled source domain to an unlabeled target domain, aiming to alleviate the constraints of data collection and annotation in training deep neural networks [1–5]. However, the application of UDA techniques is limited in some scenarios like personal medical records, where access to source data is restricted due to privacy-preserving policies. To solve this problem, source-free domain adaptation (SFDA) methods [6–8] were introduced recently, which assume the availability of only unlabeled target domain data and a pre-trained source model during adaptation. Although SFDA methods reduce the possibility of privacy leaks by using the pre-trained source model instead of source data, certain generation techniques like [9, 10] can potentially reconstruct the source data by learning from the source model. Compared to UDA and SFDA settings, domain adaptation of black-box predictors (DABP) provides better data privacy protection with more flexible portability, which adapts a model using only the unlabeled target data and a black-box predictor trained on the source domain, *e.g.*, an API service in the cloud [11]. *As demonstrated in Appendix A, we present a detailed exposition of the respective processes and the differences among UDA, SFDA, and DABP.*

---

[*] *Corresponding author.*
[†] *Both authors contributed equally to this work.*

39th Conference on Neural Information Processing Systems (NeurIPS 2025).

Recently, some researchers have paid attention to DABP, and the proposed methods [8, 11, 13–15] have made modest contributions. Early works [16–18] employ self-training learning with the pseudo-labelling techniques to subdivide target noisy labels and select reliable samples for model training. These self-training methods initially absorb useful target data and adapt well, but eventually forget target knowledge due to the accumulation of pseudo-label noise, leading to model collapse [13]. Recent proposed methods [11, 14, 15] distill the source knowledge and then fine-tune the distilled model to further fit the target domain through a two-step (distillation & fine-tuning) process. The core of the two-step process is to use high-reliability samples or the most reliable estimates to suppress negative information about low-reliability samples during training. However, this suppression leads to some problems: i) Excessive focus on aggregating reliable samples or estimates leads to premature overfitting, thereby limiting the model's ability to generalize effectively to the target data; ii) For low-reliability samples or estimates, only simple processing is applied, without any effective methods specifically to extract their useful information. Figure 1 illustrates the decline in model generalization due to overfitting in the previous methods. In addition, the fine-tuning process involves additional computational costs compared to pseudo-labeling methods.

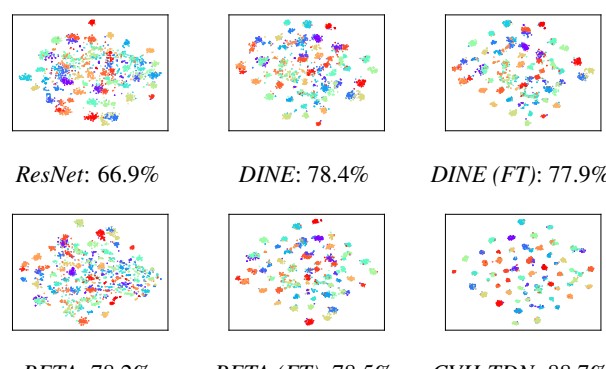

*ResNet*: 66.9%    *DINE*: 78.4%    *DINE (FT)*: 77.9%

*BETA*: 78.2%    *BETA (FT)*: 78.5%    *CVH-TDN*: 88.7%

Figure 1: The feature visualizations on the Office-Home (A→P) using t-SNE [12]. The points represent target samples and the different colors correspond to their ground-truth classes. Previous DABP methods struggle with handling a few low-reliability samples effectively. Even if the fine-tuning (FT) process is used to force the aggregation of low-reliability samples, the model still struggles to judge, leading to confusion between similar classes and a risk of degraded or stagnant performance. CVH-TDN solves this problem and improves model generalization without the FT process.

In the fields of pathology, Thalamus-driven Decoupling Network (TDN) [19] observes that abnormal synchronous activity between the thalamus and cortex, when combined with memory confusion, causes functional thalamic hallucinations and cognitive impairments. Therefore, TDN analyzes human responses from lesion studies and summarizes an explanatory framework for synchronous hallucinatory attention. In neuroscience, [20] further interprets this hallucination framework from the perspective of human attention, dividing the process of TDN into three parts: hallucination manifestations, human responses, and lesion treatment. In DABP, the black-box predictor resembles a cognitively impaired "person" who possesses only partial knowledge and is prone to errors. As a result, the target model's learning process is inevitably influenced by this noisy guidance from the black-box predictor. To address this issue, we draw inspiration from neuroscience, where such conditions are categorized as hallucination disorders: situations in which useful and interfering information are blended, preventing the brain to make reliable decisions.

Inspired by TDN, we propose a novel spatial learning method, named *Controlled Visual Hallucination via Thalamus-driven Decoupling Network* (CVH-TDN), to address the existing DABP problems. CVH-TDN builds on TDN's exploration of cognition-hallucination relationships to drive the exploration of the relationship between samples and hallucinations in the feature space. As shown in Figure 2, drawing on the connection between hallucination and brain cognition revealed by TDN, we divide CVH-TDN into three main modules: Hallucination Generation (HG), Hallucination Alignment (HA), and Hallucination Calibration (HC). In HG, considering that the generation of hallucinations is a random process that deepens with cognition, we design a progressive feature masking mechanism to control the direction of the generation process in feature space. HG generates some controlled masking in the areas specified by our strategy. With the increase of the prediction reliability, the masking degree increases in the specific areas of images that the model focuses on. In HA, based on the connections between real cognition and hallucination, we use contrast learning to explore the spatial relationship between the target sample and the corresponding generated image. In HC, we design a hierarchical calibration method that aggregates low-reliability samples with similar features

toward the nearest high-reliability samples and performs hierarchical extension based on the feature spatial similarity among samples. The calibration corresponds to the cognition-driven hallucination understanding, where random but controllable hallucinations are guided by cognitively significant objects for calibration. Experimental results demonstrate that our approach significantly outperforms the previous state-of-the-art methods on four benchmark datasets.

Our contributions can be summarized as follows: (1) We observe the weaknesses of existing DABP methods and address them by proposing a novel method, called CVH-TDN, that significantly enhances the reasoning ability of model and discrimination capacity of classes. (2) Based on the relationship between hallucination and cognition, CVH-TDN contains three parts: Hallucination Generation, Hallucination Alignment, and Hallucination Calibration, aiming to explore the

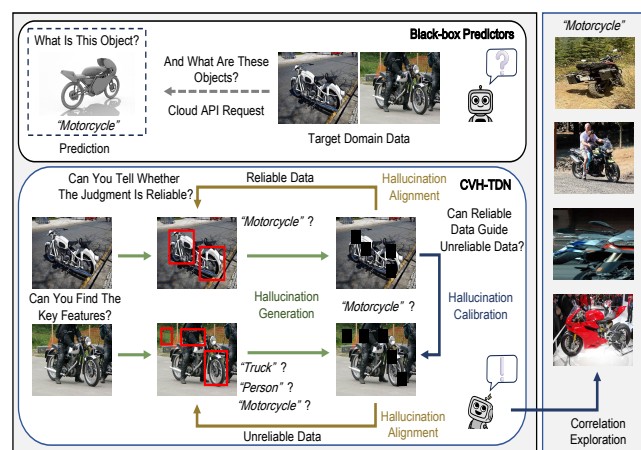

Figure 2: Conceptual figure of CVH-TDN. The black-box predictors resemble agents with prior knowledge but lack the ability to perform targeted discrimination. HG controls mask formation by modeling the location where hallucinations are pathologically generated, driven by the key cognitive impairments observed in TDN. HA improves feature discrimination by simulating how humans deal with cognitive impairments. HC draws on neurotherapeutic principles to guide unreliable data through reasoning using reliable feature representations.

spatial relationships between samples and hallucinations. (3) We perform extensive experiments to verify the effectiveness of CVH-TDN, and the results show that it achieves SOTA performance on four benchmarks. Moreover, we demonstrate the effectiveness of each component of our model and discuss the relationship between them through a large number of ablation experiments.

## 2    Related Works

**Domain Adaptation of Black-box Predictors.** With the development of generative adversarial networks (GAN), GAN-based white-box attacks are becoming more and more mature recently, requiring source models from which original source data can be recovered [13]. In early work, [18] proposed the DABP, which has almost no risk of privacy leakage. Recent method DINE [11] has found that a two-step (distillation & fine-tuning) process can encourage source-target class alignment, which distills knowledge and then fine-tunes the distilled model to adapt to the target distribution. DINE implicitly leverages the confirmed reliable knowledge from the distillation stage to cluster unreliable samples towards reliable ones during the fine-tuning process. Based on the two-step process, [14] establishes a threshold to explicitly divide the high- and low-reliability subdomains and further aligns the distribution discrepancy between the two subdomains. Based on [11, 14], [15] introduces neighborhood clustering to prevent the model from forgetting minority classes. Another line of work [13] mimics human memory to mitigate the "forgetting" problem. This work distinguishes useful information and irrelevant information, primarily focusing on learning useful samples while handling irrelevant ones on the fly. Although these methods have contributed, they use high-reliability samples to suppress the negative information from low-reliability samples, which leads to the effective information of low-reliability samples being ignored. Different from the above methods, CVH-TDN employs the sample spatial similarity to replace the targeted sample information suppression. *More recent DA works and the advantages of CVH-TDN are presented in Appendix A.*

**Visual Hallucination and Thalamus-Driven Decoupling Network.** In computer vision, some works [21–23] have found that deep features correspondence information can be discovered by jointly learning the input data and hallucinatory data with signature features. However, they are all unidirectional exploration techniques in which hallucinations are only used to reconstruct the original samples. In the absence of hallucinatory data, revealing connections for deep feature correspondences among samples is unexplored. In the field of neuroscience, Thalamus-driven Decoupling Network (TDN) [19] reveals the process and principle of human hallucinations, and mimics the effects of

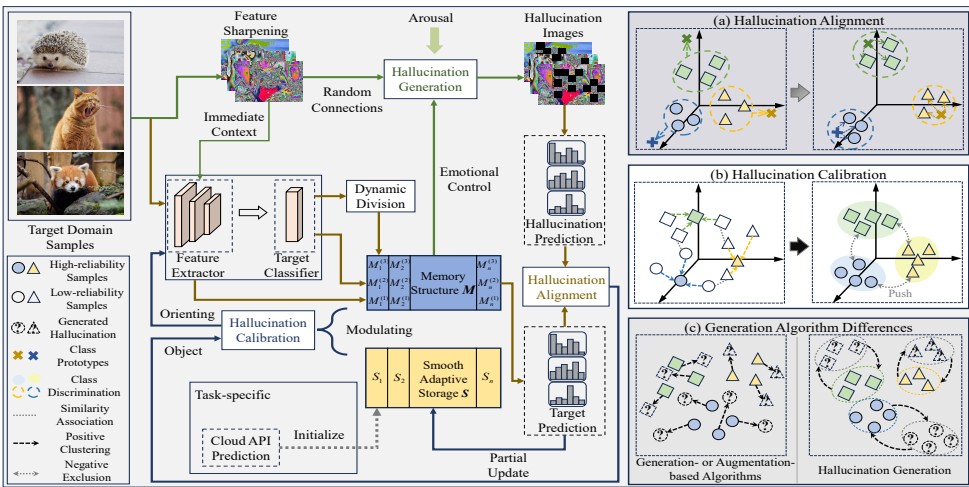

Figure 3: Overview of the completed training period of CVH-TDN. The processing flow of each module is indicated by arrows of the same color. Before training, we upload target data and obtain hard predictions from the cloud API. Feature extractor controls masking direction by evaluating knowledge from sharpened images, and the difference between the hallucination generation and other methods is shown in (c). In the hallucination alignment, we reduce the difference between samples and corresponding hallucinations by bidirectional alignment, as shown in (a). As shown in (b), we adopt hierarchical learning with the dynamic division for different types of samples based on spatial information in the hallucination calibration.

hallucinations when losing consciousness by interrupting information flow. Based on TDN, we fill the gap in uncovering the correspondence deep features between hallucination and reality without hallucinated images. Unlike generation- or augmentation-based methods [24–26], our method generates controlled hallucination masking in the model interest area of the image without additional networks, which avoids model degradation by controlling the moving direction of the cluster center on the feature space.

## 3   Proposed Method

The whole overview and the entire process of our method are illustrated in Figure 3. First, we assume that a target dataset $D_t = \{(x_i)\}_{i=1}^{N_t}$ containing $N_t$ unlabeled samples, where $x_i \in X$ and $X$ is the image set. According to the DABP setting, we upload the target dataset $D_t$ to black-box predictors (*i.e.*, cloud API services) and obtain the hard predictions $P_s$ of target samples from a source model stored in the predictors, where the source model trained by source dataset $D_s = \{(x_i^s, y_i^s)\}_{i=1}^{N_s}$ consisting of $N_s$ labeled samples. After obtaining the hard predictions, we introduce target data $x$ and initialize them into controlled generation version data $x^g$. With the increase of model training, $x^g$ are transformed from the unmasked or slightly masked version images to the masked version images in specific areas that the model focuses on. Our model $\mathcal{M}_\theta$ is parameterized by $\theta$ and composed of two components, namely, a feature extractor $F$ and a prediction classifier $C$. The feature extractor is denoted by $F : X \rightarrow Z \in \mathbb{R}^K$, where $K$ is the dimension of the feature space, and $Z$ is the $K$-dimensional transitional output from the bottleneck layer. The prediction classifier is denoted by $C : Z \rightarrow Y \in \mathbb{R}^N$, where $N$ is the number of predicted classes and $Y$ is the classification prediction output.

### 3.1   Hallucination Generation

As revealed by TDN, the process of hallucination generation is jointly determined by controllable factors (*e.g.*, context, emotion, memory, *etc.*) and uncontrollable factors (random connections). To generate exploitable hallucinations, we utilize the local areas and their adjacent context information to generate some specific masking blocks by evaluating the global image and comparing it with each local area. The strategy introduces a semi-automatic generation way that combines targeted generation and random masking. Specifically, we first set the masking block to a square of patch size

$p$ and the class location weights $\alpha_k^n$ to connect the $k$-th dimensional feature with the $n$-th class. The class location weights $\alpha_k^n$ can be defined as follows:

$$\alpha_k^n = \frac{1}{h \times w} \sum_{i=1}^{h} \sum_{j=1}^{w} \frac{\partial Y^n}{\partial z_{i,j}^k}, \tag{1}$$

where $h$ is the number obtained through dividing the image height $H$ by the masking height $p$, $i \in [1, ..., h = \frac{H}{p}]$; $W$ is the image width and divided by the masking width $p$ to obtain the number $w$, $j \in [1, ..., w = \frac{W}{p}]$; the patch size $p$ of masking block is set to $\max\left(\min\left(H, W\right)/32, 8\right)$; $k$ is the index of the feature space dimension $K$, $z^k$ is the $k$-th dimensional feature of the transitional outputs $Z$, $k \in [1, ..., K]$; $n$ is the class index and $Y^n$ represents the classification prediction for the $n$-th class, $n \in [1, ..., N]$. And $\frac{\partial Y^n}{\partial z_{i,j}^k}$ is the gradient information obtained by backpropagation of $n$-th class on the $k$-th dimensional feature.

Then, we perform a weighted combination of forward activation maps. The attention computing process is defined as follows:

$$Att_{i,j} = max\left(Tanh\left(\sum_n \sum_k a_k^n z_{i,j}^k / \sum_h \sum_w \sum_k z_{h,w}^k\right), 0\right), \tag{2}$$

where $a_k^n z_{i,j}^k$ is the linear combination of maps; $max\left(Tanh(\cdot)\right)$ is used to suppress negative pixels from other categories that the model is not interested in. $Z$ is the $K$-dimensional transitional output from the bottleneck layer, which stores $z_{h,w}^k$ as a matrix, where each $z_{h,w}^k$ is a scalar at the corresponding position in $Z$. If there is no $max\left(Tanh(\cdot)\right)$, attention values sometimes highlight not only the desired class but also other elements, and thus perform worse in feature localization. We only randomly generate masking where the model is of interest, and the masking generated formula is as follows:

$$Mask_{\substack{ip+1:(i+1)p, \\ jp+1:(j+1)p}} = [Att_{i,j} > r \ \& \ u_{i,j} > r_2], \tag{3}$$

where $r$ is a hyperparameter that controls the arousal ratio between regions of model interest and non-interest; $u_{i,j}$ is a random number, $u_{i,j} \sim U\left(0, 1\right)$; $r_2$ is the uncontrollable masking ratio and is set to 0.5. Subsequently, we simply sharpen the features of data $x$ and combine the sharpened data $\widetilde{x}$ with the generated masks, as follows:

$$x^g = Mask \odot \widetilde{x}, \tag{4}$$

where $\odot$ is the element-wise multiplication symbol.

In the initial training, due to the lack of prior model training, the model attention for the entire image is distributed and unreliable. At this stage, according to Eq. (3), $x^g$ represent the unmasked or slightly masked hallucination version of the image. With increased model discrimination, the generation of $x^g$ controlled by the model attention generates controllable masking in the area of model interest.

### 3.2 Hallucination Alignment

In the hallucination state model, predictions of sensory-level data assume crucial roles. These predictions reflect prior information about what could occur in the upcoming hallucination environment: *i.e.*, what is expected to be seen in that context. The focus of this work is how to efficiently explore sensory level data (*i.e.*, the model's immediate predictions and importance assessment of the original samples). For this, we first introduce a permanent memory storage $M$, which stores the global predicted probabilities and mimics a memory structure that stores and processes sensory-level data. With each iteration, $M$ overwrites the probabilities for the current batch samples. The storage $M$ is the key container of our method, which controls the movement direction of the samples on the feature space by acquiring the sample state immediately and can be expressed as follows:

$$M = \{(Hier(x_i), norm(Z_i), Softmax(Y_i))\}_{i=1}^{N_t} = \{(M_i^{(1)}, M_i^{(2)}, M_i^{(3)})\}_{i=1}^{N_t}, \tag{5}$$

where $M^{(k)}$ corresponds to the $k$-th subset in $M$; $norm(\cdot)$ denotes the normalization function; $norm(Z_i)$ represents the spatial features of the samples, which is used to calculate the similarity between the samples; $Hier(\cdot)$ is the hierarchical strategy to judge the current sample is reliable $R$ or unreliable $UR$. The judgment process can be expressed as:

$$Dif(Y_i) = TopK(Y_i, 1) - TopK(Y_i, 2), \tag{6}$$

$$\tau_1 = \frac{\lambda}{N_t} \sum_{i=1}^{N_t} Dif\left(Y_i\right), \tau_2 = \frac{\lambda}{N_t} \sum_{i=1}^{N_t} TopK\left(M_i^{(3)}, 1\right), \tag{7}$$

where $TopK\left(\cdot, j\right)$ defines the $j$-th largest value in the corresponding tensor; the threshold of $\tau_1$ is to filter samples that are at the boundary of similar classes in the feature space, and the threshold of $\tau_2$ is to filter samples that are judged unreliable in the current batch. Therefore,

$$Hier(x_i) = \begin{cases} R_i, \ if \ Dif\left(Y_i\right) > \tau_1 \ \& TopK\left(M_i^{(3)}, 1\right) > \tau_2 \\ UR_i, \ otherwise \end{cases}, \tag{8}$$

where $\lambda$ is a hyperparameter that controls the ratio between reliable and unreliable samples in a mini-batch. The combination of $\tau_1$ and $\tau_2$ constitutes our hierarchical strategy.

Then, we design a bidirectional weight module $w$ to assign different weights by pairing features with different similarities between the original image and the corresponding generated image. The module can be defined as follows:

$$w = \exp(\log(sim(M^{(2)}, norm(Z^g)))), \tag{9}$$

where $Z^g$ is denoted the transitional output of the generated image $x^g$; $sim(\cdot)$ is the operation of calculating cosine similarity. The hallucination alignment deploys contrastive learning to explore the spatial relationship, its process of calculating the loss is as follows:

$$\mathcal{L}_{HA}^{forward} = -w \times \log(sim(M^{(3)}, \sigma(Y^g))), \tag{10}$$

$$\mathcal{L}_{HA}^{back} = -\log(1 - sim(M^{(3)}, \sigma(Y^g))), \tag{11}$$

$$\mathcal{L}_{HA} = \mathcal{L}_{HA}^{forward} + \mathcal{L}_{HA}^{back}, \tag{12}$$

where $Y^g$ is the prediction of the generated data $x^g$ and $\sigma(\cdot)$ is the Softmax. $\mathcal{L}_{HA}$ is composed of bidirectional losses: $\mathcal{L}_{HA}^{forward}$ encourages samples whose features are similar to be aligned with the generated version in the feature space; $\mathcal{L}_{HA}^{back}$ calibrates $\mathcal{L}_{HA}^{forward}$ and reduces the noise caused by common features to prevent overfitting of the model.

### 3.3 Hallucination Calibration

Previous DABP methods [13–15] maintained a fixed momentum parameter when updating the adaptive label smoothing [11], which is considered to perform better than the original model output $Y$ and can be expressed as follows:

$$S(x_i) = \begin{cases} \frac{1}{N} \sum_{j=1}^{N} AdaLS(P_s^{i,(n)}), beginning \\ \mu S(x_i) + (1 - \mu)Y_i, otherwise \end{cases}, \tag{13}$$

where $AdaLS(\cdot)$ is the calculation of the adaptive label smoothing; $P_s^{i,(n)}$ is the hard prediction for $(n)$-th class of the $i$-th sample used as input from the black-box predictors before the training; $\mu$ is the momentum hyperparameter that holds a fixed value in these methods. In this study, we found that this operation has some problems: i) The initial hard predictions from the black-box predictors are more robust than the training model, but this operation pays more attention to the early training model, resulting in the accumulated noise in the early training and affecting the subsequent training; ii) During the mid-to-end training stages, the model discriminant ability and the output reliability are enhanced, but the update variable is fixed, which limits the efficiency of extracting feature knowledge. Different from them, we replace $\mu$ with dynamic variable $\tilde{\mu} = \begin{cases} \tilde{\mu}_s, begining \\ \tilde{\mu} - Iter \times (\tilde{\mu} - \mu)/Iter\_total, other \end{cases}$ to update $S(x_i)$ that is similar to the brain cognitive process, where $Iter$ is the number of iterations and $\tilde{\mu}_s$ is to initialize information extraction. *The process details are described in Appendix F.*

Then, the hallucination calibration is divided into two parts: i) Hierarchical sample calibration (Hsc) and ii) Common feature exclusion (Cfe). The former is to calibrate samples with similar signature features, and the latter is to separate ambiguous samples with similar common features. In Hsc, the high-reliability samples are distinguished by the previous hierarchical strategy to cluster the low-reliability samples that are close in spatial distance. The computation can be expressed as:

$$\mathcal{L}_{HC}^{Hsc} = \frac{1}{N_t} \sum_{i=1}^{N_t} D_{KL}(M_i^{(3)} || \{(M_{sim2(M_i^{(2)}, M_j^{(2)}, \phi)}^{(3)})\}^{M_j^{(1)} = R}), j \in [1, 2, ..., \phi], \tag{14}$$

Table 1: Accuracies (%) on the *Office-Home* using the ResNet-50 backbone. ▢ denotes the hard task whose source-only accuracy is below 65%. **H.Mean** denotes the average accuracy of hard tasks. The top-performing DABP methods are highlighted in bold.

| Method | DABP | A→C | A→P | A→R | C→A | C→P | C→R | P→A | P→C | P→R | R→A | R→C | R→P | Mean | H.Mean |
|---|---|---|---|---|---|---|---|---|---|---|---|---|---|---|---|
| Source-only | − | 44.1 | 66.9 | 74.2 | 54.5 | 63.3 | 66.1 | 52.8 | 41.2 | 73.2 | 66.1 | 46.7 | 77.5 | 60.6 | 50.4 |
| CDAN | × | 52.0 | 68.6 | 76.1 | 58.0 | 70.3 | 70.2 | 58.6 | 50.2 | 77.6 | 72.2 | 59.3 | 81.9 | 66.3 | 58.1 |
| CST | × | 59.0 | 79.6 | 83.4 | 68.4 | 77.1 | 76.7 | 68.9 | 56.4 | 83.0 | 75.3 | 62.2 | 85.1 | 73.0 | 65.3 |
| HMA | × | 60.6 | 79.1 | 82.9 | 68.9 | 77.5 | 79.3 | 69.1 | 55.9 | 83.5 | 74.6 | 62.3 | 84.4 | 73.2 | 65.7 |
| LNL-KL | ✓ | 49.0 | 71.5 | 77.1 | 59.0 | 68.7 | 72.9 | 56.4 | 46.9 | 76.6 | 66.2 | 52.3 | 79.1 | 64.6 | 55.4 |
| HD-SHOT | ✓ | 48.6 | 72.8 | 77.0 | 60.7 | 70.0 | 73.2 | 56.6 | 47.0 | 76.7 | 67.5 | 52.6 | 80.2 | 65.3 | 55.9 |
| SD-SHOT | ✓ | 50.1 | 75.0 | 78.8 | 63.2 | 72.9 | 76.4 | 60.0 | 48.0 | 79.4 | 69.2 | 54.2 | 81.6 | 67.4 | 58.1 |
| DivideMix | ✓ | 51.7 | 74.7 | 78.5 | 61.8 | 72.4 | 73.3 | 59.8 | 48.0 | 82.9 | 68.0 | 56.4 | 81.6 | 67.4 | 58.4 |
| DINE | ✓ | 52.2 | 78.4 | 81.3 | 65.3 | 76.6 | 78.7 | 62.7 | 49.6 | 82.2 | 69.8 | 55.8 | 84.2 | 69.7 | 60.4 |
| BiMem | ✓ | 54.5 | 78.8 | 81.4 | 66.7 | 78.7 | 79.6 | 65.9 | 53.6 | 82.3 | 73.6 | 57.8 | 84.9 | 71.5 | 62.9 |
| BETA | ✓ | 57.2 | 78.5 | 82.1 | 68.0 | 78.6 | 79.7 | 67.5 | 56.0 | 83.0 | 71.9 | 58.9 | 84.2 | 72.1 | 64.4 |
| RFC | ✓ | 57.4 | 80.0 | 82.8 | 67.0 | 80.6 | 80.2 | 68.3 | 57.8 | 82.8 | 72.8 | 59.3 | 85.9 | 72.9 | 65.1 |
| SEAL | ✓ | 58.5 | 81.4 | **84.7** | **71.7** | 80.4 | 82.1 | **72.2** | 54.3 | **86.0** | **76.2** | 60.6 | 86.3 | 74.5 | 66.3 |
| **CVH-TDN** | ✓ | **71.7** | **88.7** | 83.3 | 69.7 | **86.1** | **83.3** | 70.2 | **68.9** | 83.7 | 73.8 | **72.6** | **91.3** | **78.6** | **73.2** |

where $D_{KL}(\cdot)$ is the Kullback-Leibler divergence; $sim2(\cdot, \phi)$ is the index set of the first-$\phi$ similar samples by calculating cosine similarity.

In Cfe, we introduce a regularization loss to enhance the difference in the representation of different samples in the feature space, and the generated hallucination image is forcibly aligned with the original image to guide the separation of different classes of samples. The regularization loss can be expressed as:

$$\mathcal{L}_{HC}^{Cfe} = \frac{1}{N_R} \sum_{i=1}^{N_R} (\overline{Y_i} \log \sigma(Y_i^g) + M_i^{(3)^T} \times \sum_{j=1, i \neq j}^{N_R} M_j^{(3)}), \tag{15}$$

where $\overline{Y_i} = \arg\max Y_i$ is a hard label predicted by the training model, $N_R$ is the number of high-reliability samples, and $T$ is the transpose operation. This process requires the gradient to be updated separately, so that the class differences can be improved through common feature information without being affected by the information of individual samples. The $L_{HC}$ is denoted by:

$$\mathcal{L}_{HC} = \mathcal{L}_{HC}^{Hsc} + \mathcal{L}_{HC}^{Cfe}. \tag{16}$$

According to [11], the training model is affected by negative information from low-reliability samples in early training, which leads to the decline of model discrimination ability. In the hallucination calibration, we make low-reliability samples unaffected by $\mathcal{L}_{HC}^{Cfe}$ initially, and use $\mathcal{L}_{HC}^{Hsc}$ to explore the spatial similarity among samples. In the middle of training, driven by the hallucination alignment, all type-$UR$ samples will become samples of type $R$, where $N_R = N_t$. $\mathcal{L}_{HC}$ discovers the spatial information with deep feature correspondence by jointly learning hallucinations and similar samples with signature features.

### 3.4 Overall Objective Function

Following previous methods [11, 13, 14], the conditional self-regularization is adopted as the task-specific loss to complete the DABP task. The task-specific loss can be expressed as:

$$L_{task} = \mathbb{E}_{x \in X} D_{KL}(Y||S(x)). \tag{17}$$

Therefore, the overall objective loss of CVH-TDN can be expressed as:

$$\mathcal{L}_{total} = \mathcal{L}_{HA} + \mathcal{L}_{HC} + \mathcal{L}_{task}. \tag{18}$$

*To explain why CVH-TDN works effectively and why it contributes to DABP, we derive an error bound through theoretical analysis in Appendix B. The whole training process is shown in Appendix C.*

## 4 Experiments

**Implementation Details.** We evaluated our method on four standard benchmark datasets, including *Office-31* [27], *Office-Home* [28], *VisDA-17* [29], and *DomainNet* [30]. Our method is implemented based on the PyTorch and train the model on a machine with an NVIDIA GeForce RTX4090 GPU.

Table 2: Accuracies (%) on the *VisDA-17* using the ResNet-101 backbone.

| Method | DABP | plane | bike | bus | car | horse | knife | mcycle | person | plant | sktbrd | train | truck | Mean | H.Mean |
|---|---|---|---|---|---|---|---|---|---|---|---|---|---|---|---|
| Source-only | − | 64.3 | 24.6 | 47.9 | 75.3 | 69.6 | 8.5 | 79.0 | 31.6 | 64.4 | 31.0 | 81.4 | 9.2 | 48.9 | 35.2 |
| MCC | × | 88.7 | 80.3 | 80.5 | 71.5 | 90.1 | 93.2 | 85.0 | 71.6 | 89.4 | 73.8 | 85.0 | 36.9 | 78.8 | 76.8 |
| HMA | × | 97.6 | 88.4 | 84.3 | 76.0 | 98.4 | 97.1 | 91.3 | 81.4 | 97.0 | 96.7 | 88.8 | 60.7 | 88.1 | 87.9 |
| COT | × | 98.2 | 89.4 | 87.6 | 82.3 | 98.0 | 97.2 | 96.4 | 86.2 | 98.3 | 92.6 | 92.2 | 58.1 | 89.7 | 88.5 |
| LNL-KL | ✓ | 82.7 | 83.4 | 76.7 | 44.9 | 90.9 | 38.5 | 78.4 | 71.6 | 82.4 | 80.3 | 82.9 | 50.4 | 71.9 | 70.8 |
| HD-SHOT | ✓ | 75.8 | 85.8 | 78.0 | 43.1 | 92.0 | 41.0 | 79.9 | 78.1 | 84.2 | 86.4 | 81.0 | 65.5 | 74.2 | 74.4 |
| SD-SHOT | ✓ | 79.1 | 85.8 | 77.2 | 43.4 | 91.6 | 41.0 | 80.0 | 78.3 | 84.7 | 86.8 | 81.1 | 65.1 | 74.5 | 74.8 |
| DINE | ✓ | 81.4 | 86.7 | 77.9 | 55.1 | 92.2 | 34.6 | 80.8 | 79.9 | 87.3 | 87.9 | 84.3 | 58.7 | 75.6 | 74.3 |
| BETA | ✓ | 94.9 | 90.2 | 85.4 | 61.1 | 95.5 | 93.1 | 85.0 | 83.8 | 92.9 | 91.9 | 91.1 | 55.0 | 85.1 | 85.9 |
| RFC | ✓ | 95.6 | 89.7 | 87.8 | 75.8 | 96.5 | **96.5** | 90.4 | 82.8 | **96.0** | 70.0 | 85.7 | 55.1 | 85.2 | 84.2 |
| SEAL | ✓ | **97.9** | **92.2** | **88.0** | 73.5 | **97.1** | 96.1 | 92.4 | 85.7 | 93.9 | **95.6** | 91.2 | 66.4 | 89.2 | 89.5 |
| **CVH-TDN** | ✓ | 96.9 | 91.6 | 87.3 | **83.6** | 97.0 | 95.8 | **92.7** | **90.6** | 95.9 | 95.5 | **92.6** | **67.2** | **90.6** | **90.1** |

To make fair comparisons, we follow [11, 31] to select the ResNet [1] pre-trained on ImageNet [32] as the backbone in all experiments, where ResNet-50 is used for *Office-31*, *Office-Home*, and *DomainNet*, while ResNet-101 is used for *VisDA-17*. In our experiments, we adopt SGD optimizer with the weight decay 1e-3, the momentum 0.9, the feature extractor learning rate 1e-4, and the classifier learning rate 1e-3. The bottleneck size is set to 256 and the batch size is set to 64. Following previous SOTA [14], the number of warm-ups is set to 3, and both overall-average accuracy (**Mean**) and on-hard task (whose source-only accuracies are below 65%) average accuracy (**H.Mean**) are reported. *Specific dataset details are presented in Appendix D.*

**Comparison Methods.** To evaluate our method, we choose several related UDA and DABP works for comparison. For UDA, we compare to CDAN [33], MCC [34], HMA [35], and CST [36]. Meanwhile, we compare our CVH-TDN with previous SOTA DABP methods, including LNL-KL [16], HD-SHOT [8], SD-SHOT [8], DivideMix [37], DINE [11], BiMem [13], BETA [14], RFC [15], and SEAL [31]. To ensure fair comparisons, we follow the training protocol, learning strategy, and network architecture for the source domain as specified in DINE. The results of all comparison methods are obtained from the original papers, their associated codebases, or follow-up work.

Table 3: Accuracies (%) on the *Office-31* using the ResNet-50 backbone.

| Method | DABP | A→D | A→W | D→A | D→W | W→A | W→D | Mean | H.Mean |
|---|---|---|---|---|---|---|---|---|---|
| Source-only | − | 79.9 | 76.6 | 56.4 | 92.8 | 60.9 | 98.5 | 77.5 | 58.7 |
| MCC | × | 95.6 | 95.4 | 72.6 | 98.6 | 73.9 | 100 | 89.4 | 73.3 |
| HMA | × | 95.8 | 95.1 | 79.3 | 99.3 | 77.6 | 100 | 91.2 | 78.5 |
| LNL-KL | ✓ | 89.4 | 86.8 | 65.1 | 94.8 | 67.1 | 98.7 | 83.6 | 66.1 |
| HD-SHOT | ✓ | 86.5 | 83.1 | 66.1 | 95.1 | 68.9 | 98.7 | 83.0 | 67.5 |
| SD-SHOT | ✓ | 89.2 | 83.7 | 67.9 | 95.3 | 71.1 | 97.1 | 84.1 | 69.5 |
| DINE | ✓ | 91.6 | 86.8 | 72.2 | 96.2 | 73.3 | 98.6 | 86.4 | 72.8 |
| BiMem | ✓ | 92.8 | 88.2 | 73.9 | 96.8 | 75.3 | 99.4 | 87.7 | 74.6 |
| BETA | ✓ | 93.6 | 88.3 | 76.1 | 95.5 | 76.5 | 99.0 | 88.2 | 76.3 |
| RFC | ✓ | 94.4 | **93.0** | 76.7 | 95.6 | 77.5 | 98.1 | 89.2 | 77.1 |
| SEAL | ✓ | 95.1 | 88.3 | **77.6** | 96.0 | 76.7 | 99.3 | 88.8 | 77.2 |
| **CVH-TDN** | ✓ | **96.4** | 92.8 | 75.6 | **98.9** | **81.0** | **99.6** | **90.7** | **78.3** |

Table 4: Results with different values of hyperparameter $\lambda$ on *Office-31* with ResNet-50 backbone.

| $\lambda$ | A→D | A→W | D→A | D→W | W→A | W→D | Mean |
|---|---|---|---|---|---|---|---|
| 0 | 92.8 | 91.3 | 73.2 | 98.5 | 76.8 | 99.6 | 88.7 |
| 10% | **96.4** | 92.8 | **75.6** | 98.9 | **81.0** | 99.6 | **90.7** |
| 20% | 95.8 | **93.2** | 73.8 | 98.9 | 77.8 | 99.6 | 89.8 |
| 30% | 95.4 | 93.0 | 72.4 | **99.0** | 74.4 | **99.8** | 89.0 |

Table 5: Results of ablation study on the *Office-31* and *VisDA-17* datasets.

| Loss | | | Office | | | | | | | VisDA |
|---|---|---|---|---|---|---|---|---|---|---|
| $\mathcal{L}_{HA}$ | $\mathcal{L}_{HC}$ | $HG$ | A→D | A→W | D→A | D→W | W→A | W→D | Mean | Mean |
| Source only | | | 79.9 | 76.6 | 56.4 | 92.8 | 60.9 | 98.5 | 77.5 | 48.9 |
| ✓ | | | 89.3 | 86.4 | 73.6 | 96.5 | 74.4 | 99.0 | 86.5 | 75.7 |
| | ✓ | | 89.8 | 86.8 | 74.5 | 98.5 | 75.2 | 99.6 | 87.4 | 78.1 |
| ✓ | ✓ | | 94.4 | 90.4 | 74.7 | 98.9 | 80.1 | 99.8 | 89.7 | 85.1 |
| ✓ | | ✓ | 93.8 | 92.1 | 74.5 | 98.6 | 77.1 | **99.9** | 89.3 | 83.5 |
| | ✓ | ✓ | 92.9 | 90.2 | 73.9 | 98.6 | 78.9 | 99.4 | 89.0 | 82.5 |
| ✓ | ✓ | ✓ | **96.4** | **92.8** | 75.6 | **98.9** | **81.0** | 99.6 | **90.7** | **90.6** |
| $\mathcal{L}_{HA}$ | $\mathcal{L}_{HC}^{Hsc}$ | $\mathcal{L}_{HC}^{Cfe}$ | | | | | | | | |
| ✓ | ✓ | | 94.5 | 91.9 | 75.8 | 98.7 | 79.7 | 99.6 | 90.0 | 88.7 |
| ✓ | | ✓ | 94.7 | 92.3 | **76.1** | 98.7 | 80.1 | 99.6 | 90.3 | 87.1 |

**Results.** In Tables 1, 2, 3, and *Appendix E*, the results show that CVH-TDN significantly outperforms the previous SOTA methods on all four benchmarks. For average accuracy, CVH-TDN outperforms the previous best results by 4.1%, 1.6%, 1.5%, and 1.4% on *Office-Home*, *DomainNet*, *Office-31*, and *VisDA-17*, respectively. For these hard tasks with lower than 65% source-only accuracy, CVH-TDN outperforms the baseline BETA by 8.8%, 1.6%, 2.0%, and 4.2% on *Office-Home*, *DomainNet*, *Office-31*, and *VisDA-17*. These results demonstrate that the exploration of sample spatial relationships

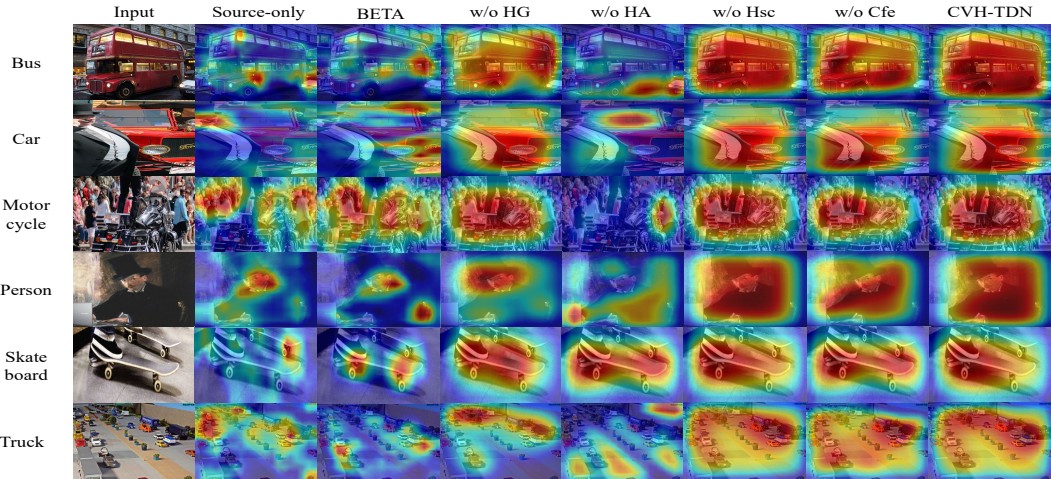

Figure 4: The visualization results of the heat map on the *VisDA-17*. Each result is reported when the best accuracy is achieved.

by controlled hallucination is more effective than the suppression of specific sample information in increasing class discriminability.

**Parameter Analysis and Comparison.** As shown in Figure 5(a), we plot the trends of prediction accuracy under different arousal ratios $r$ with all 60 epochs on the *VisDA-17* dataset. When $r$ is equal to 0.5, we observe that the best performance can be obtained. And when $r$ is equal to 0, the hallucination generation fails and all generated images are uncontrollable. These highlight that controllable hal-

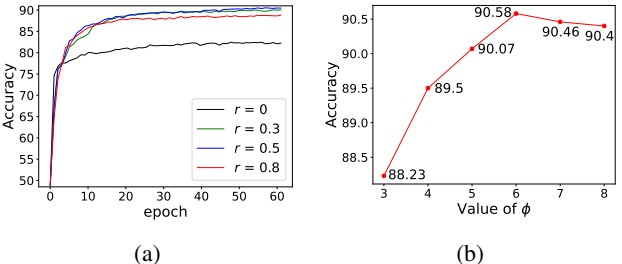

(a)  (b)

Figure 5: The accuracy trends of predictions on the *VisDA-17* dataset. (a) shows the accuracy evolution of CVH-TDN under different arousal values of $r$. (b) shows the accuracy with different values of hyperparameter $\phi$.

lucination generation is essential for improving the model reasoning ability, while uncontrollable generation can cause CVH-TDN to fail. Figure 5(b) plots the accuracy under different $\phi$ values, where $\phi$ determines the calibration influence of each high-reliability sample. As the value of $\phi$ increases, high-reliability samples tend to cluster low-reliability samples with similar signature features more capably. However, if $\phi$ is too large, common features may also be considered, leading to premature model overfitting. Table 4 analyzes the effect of hierarchical strategy under different $\lambda$ values on the *Office-31*. When $\lambda$ is equal to 0, the hierarchical strategy fails, which leads to the negative influence of the information from unreliable samples on the early learning of the model. A higher value of $\lambda$ reduces the proportion of reliable samples, leading to decreased efficiency in exploring sample spatial relationships. *More parameter analysis and comparison are shown in Appendix F.*

**Ablation Study.** We report the ablation study on the *Office-31* and *VisDA-17* in Table 5 and present the Grad-CAM [38] visualizations for the *VisDA-17* dataset in Figure 4.

As shown in Table 5, experimental results demonstrate that every part of our algorithm is necessary and indispensable. $\mathcal{L}_{HA}$ is designed to prevent overfitting of the model by controlling the original and hallucination images for bidirectional alignment, and it can effectively improve model generalization only when combined with Hallucination Generation (HG). $\mathcal{L}_{HC}$ is an efficient clustering loss, but since the effect of task-specific loss is also sample clustering, only the combined effect of the two leads to model overfitting. For this, $\mathcal{L}_{HA}$ helps $\mathcal{L}_{HC}$ to alleviate the conflict with task-specific loss. When $\mathcal{L}_{HC}$ and HG are combined, the class discrimination ability and model reasoning ability are further improved. As shown in Figure 4, both Hsc and Cfe components of $\mathcal{L}_{HC}$ contribute to improving the reasoning ability of the model by highlighting the areas that the model focuses on for a given category. When Hsc is removed, the values of the regions of interest in the model are averaged,

resulting in a lot of non-main features being captured. Both modules of $L_{HA}$ and Cfe are designed to eliminate overfitting: $L_{HA}$ is to eliminate overall sample overfitting, and Cfe is to eliminate reliable sample overfitting. When Cfe is removed, the ability to grasp key features is weakened, resulting in some key features not being captured. Combining the two, the model can extract the features in the image accurately and efficiently. In summary, $\mathcal{L}_{HA}$ and HG jointly determine the approximate region of feature extraction from the model, and $\mathcal{L}_{HC}$ optimizes features for the region of interest of the model. The results show that their combination significantly improves the model reasoning ability. *More visual studies are shown in the Appendix G.*

## 5 Conclusion

In this work, we observe some weaknesses in existing DABP methods and solve them by proposing a novel algorithm, CVH-TDN. Inspired by the Thalamus-driven Decoupling Network, CVH-TDN contains the Hallucination Generation (HG), Hallucination Alignment (HA), and Hallucination Calibration (HC) to achieve sample clustering and hallucination control by exploring the spatial relationships. HG utilizes the attention of the training model to generate exploitable hallucinations. HA explores the spatial relationship between the samples and the hallucinations generated by HG through bidirectional alignment to encourage clustering of samples with similar features. HC uses a hierarchical way to aggregate samples with similar signature features and separates ambiguous samples with common features. Experimental results demonstrate the effectiveness of spatial similarity exploration in enhancing the model reasoning and class discriminability. CVH-TDN significantly outperforms previous SOTA methods on all comparison datasets.

## Acknowledgment

This work was supported in part by the National Natural Science Foundation of China under Grant 62176162 and in part by Guangdong Basic and Applied Basic Research Foundation under Grant 2023A1515012875 and Grant 2022A1515140099.

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

# Appendix

## A. More Related Works

**Unsupervised Domain Adaptation.** UDA aims to tackle the challenge of generalizing a model trained on a large number of labeled samples from the source domain to the target domain. UDA has been extensively explored for many practical applications, including image classification [39–41], semantic segmentation [42–44], object detection [45–47], and time series forecasting [48–50]. However, UDA methods assume that both the labeled source domain and the unlabeled target domain are available at the same time, which is not available in some scenarios where the source domain data cannot be accessed during training due to privacy-preserving policies.

**Source-free Domain Adaptation (White-box Predictors).** Compared to the UDA setting, SFDA has higher privacy protection because it does not need to touch any source data. Many SFDA methods have been proposed recently [6–8, 51], which only require access to unlabeled target data and a trained source domain model during training. Most existing research on SFDA tasks is mainly based on self-training [6, 8, 51], class prototypes [8, 52], contrastive learning [7, 53], and generative models [54, 25]. Although the SFDA task has contributed to the mitigation of privacy protection issues to some extent, recent research [13] has found that exposing the details of the white-box predictive model training is quite dangerous due to certain reverse generation techniques like [9, 10].

**Setting Comparison and Method Improvement.** As shown in Figure 6, the respective processes and the differences among UDA, SFDA, and DABP are presented. Compared with SFDA, DABP provides better data privacy protection with more flexible portability, which only needs to upload target data to the cloud API and then download predictions before training. Figure 7 illustrates the advantages of CVH-TDN and the differences from previous DABP methods. Different from the previous DABP methods [11, 13–15, 31], CVH-TDN leverages sample spatial similarity instead of suppressing targeted sample information, thereby improving model generalization and class discrimination.

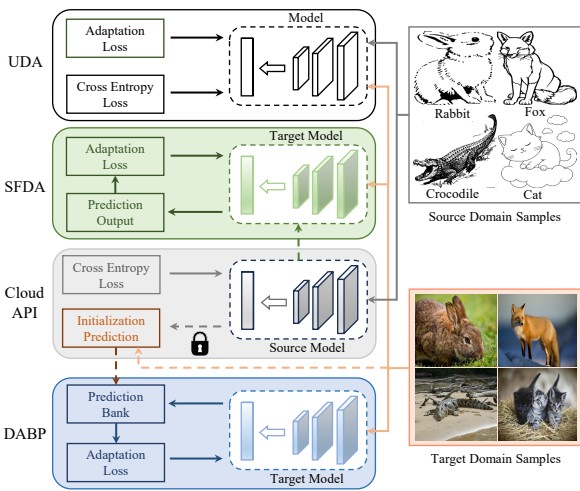

Figure 6: Illustration of the differences among UDA, SFDA, and DABP. The dotted lines in the figure indicate the operations performed by the cloud API with the source model under different settings. SFDA requires the entire source model to be obtained from the cloud API before training. DABP outperforms SFDA in data privacy protection and portability, simply requiring uploading target data to the cloud API and then downloading predictions before training.

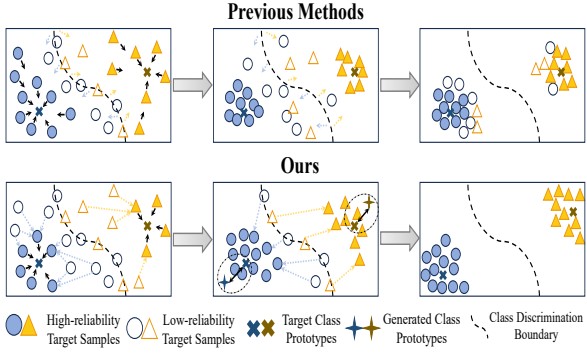

Figure 7: Comparison between existing DABP methods and CVH-TDN. Top: Existing DABP methods focus on learning high-reliability knowledge and force low-reliability sample aggregation. Bottom: CVH-TDN proposes hallucination alignment to investigate the spatial connection between samples and hallucinations and introduces hallucination calibration to explore relationships among spatially similar samples. Our algorithm enhances the capabilities of model generalization and class discrimination.

## B. Theoretical Analysis

We provide theoretical justifications grounded in the generalization bound of reasoning to clarify the working mechanism of our algorithm.

Since our algorithm is trained in an unlabeled target domain and generates the controlled samples based on our hallucination generation, we denote $x \sim D_T$ as the real sample distribution of the target domain. Based on the existing theories [55], the error of CVH-TDN can be formulated as a convex combination of the errors of the reliable subdomain $x_R \sim D_R$, the unreliable subdomain $x_{UR} \sim D_{UR}$, and the generated subdomain $x_G \sim D_G$ that represents the generated sample distribution of the reliable subdomain. And denote $y_R$, $y_{UR}$ and $\widehat{y}_R$, $\widehat{y}_{UR}$ as the true labels and the predicted labels of $x_R$ and $x_{UR}$, respectively. Denote $y_G$ as the true label and $\widehat{y}_G$ as the predicted label of $x_G$. $y_T$ and $\widehat{y}_T$ are the true labels and the predicted labels of target domain, $y_G = y_R$, $y_T = y_R + y_{UR}$, and $\widehat{y}_T = \widehat{y}_R + \widehat{y}_{UR}$. Let $H$ denote a hypothesis, which can be expressed as:

$$\epsilon_R\left(H, \widehat{y}_R\right) \leq \epsilon_G\left(H, \widehat{y}_G\right) + d_{n\Delta n}(D_R, D_G) + \varepsilon_G, \tag{19}$$

$$\epsilon_{\widehat{T}}\left(H, \widehat{y}_T\right) = \alpha\epsilon_R\left(H, \widehat{y}_R\right) + (1 - \alpha)\epsilon_{UR}\left(H, \widehat{y}_{UR}\right), \tag{20}$$

$$\epsilon_T\left(H, y_T\right) = \alpha\epsilon_R\left(H, y_R\right) + (1 - \alpha)\epsilon_{UR}\left(H, y_{UR}\right), \tag{21}$$

where $d_{n\Delta n}(D_R, D_G) = 2\sup_{H,H'\in n}|\mathbb{E}_{x_R\sim D_R}\big[H(x_R) \neq H'(x_R)\big] - \mathbb{E}_{x_G\sim D_G}\big[H(x_G) \neq H'(x_G)\big]|$; $\epsilon_G\left(H, \widehat{y}_G\right)$ is the expected error of the generated sample distribution; $\varepsilon_G = min(\epsilon_R\left(H, \widehat{y}_R\right) + \epsilon_G\left(H, \widehat{y}_G\right))$ ; $\epsilon_R\left(H, \widehat{y}_R\right)$ is the expected error of the reliable subdomain; $\alpha$ is the trade-off parameter that is controlled by $\lambda$ in Eq. (7); $\epsilon_{UR}\left(H, \widehat{y}_{UR}\right)$ is the expected error of the unreliable subdomain; $\epsilon_R\left(H, y_R\right)$ and $\epsilon_{UR}\left(H, y_{UR}\right)$ denote the oracle errors of reliable and unreliable samples, respectively.

For $\epsilon_R\left(H, \widehat{y}_R\right) \leq \epsilon_G\left(H, \widehat{y}_G\right) + d_{n\Delta n}(D_R, D_G) + \varepsilon_G$, we analyze each component in detail in this paragraph:

(1) $\epsilon_G\left(H, \widehat{y}_G\right)$ is the expected error of the generated sample distribution, which can be minimized with a cross-entropy loss in the former term of Eq. (15). With the guidance of the black-box predictors, in the initial stage of training, the distribution of the reliable samples selected by the target model in the feature space is relatively close to that of the samples in the source domain. We can obtain good training results for these samples with similar features through the black-box predictors. As the training progresses, the model gradually adapts to the distribution of the target domain. the target model continuously selects more reliable samples. Thus, $D_G$ can better learn the knowledge of these reliable samples through Eq. (15), and align the distribution of the target features through Eq. (12), so as to continuously adapt to the target domain. Therefore, $\epsilon_G\left(H, \widehat{y}_G\right)$ is small in the whole training.

(2) $\varepsilon_G$ is the shared error of the ideal joint hypothesis which is considered to be a sufficiently small constant to represent the complexity of the generated sample hypothesis space.

(3) $d_{n\Delta n}(D_R, D_G)$ depends on the expected error of the disagreement between two hypothesis on the reliable subdomain and the generated sample distribution of the reliable subdomain. In the early stages of training, it is easy to find two hypotheses that both $H$ and $H'$ correctly predict the reliable samples. As the training progresses, hallucination alignment explores the spatial relationship between the samples and the hallucinations generated by hallucination generation through bidirectional alignment to encourage clustering of samples with similar features. Hallucination alignment helps to maintain the discrimination and generalization ability of the model in the target domain. Therefore, $\mathbb{E}_{x_G\sim D_G}\big[H(x_G) \neq H'(x_G)\big]$ is always small during adaptation phase. Hallucination calibration uses a hierarchical way to aggregate samples with similar signature features and separates ambiguous samples with common features. Hallucination calibration helps to improve model reasoning ability and prevent reliable sample overfitting. The joint learning of hallucination alignment and hallucination calibration is conducted on reliable samples. As a result, $\mathbb{E}_{x_R\sim D_R}\big[H(x_R) \neq H'(x_R)\big]$ always maintaining a small value during adaptation phase.

Then, during the transition from the early stage to the middle stage of adaptation, we derive an upper bound of how the error $\epsilon_{\widehat{T}}\left(H, \widehat{y}_R\right)$ is close to $\epsilon_T\left(H, y_T\right)$ though hallucination calibration, which is the oracle error with the truth label $y_t$ of the target domain.

**Theorem 1.** Let $H$ be a hypothesis in class $n$, we have:

$$\left|\epsilon_{\widehat{T}}(H,\widehat{y}_T) - \epsilon_T(H,y_T)\right| = \left|\alpha\,\epsilon_R(H,\widehat{y}_R) + (1-\alpha)\,\epsilon_{UR}(H,\widehat{y}_{UR})\right.$$

$$\left. - \alpha\,\epsilon_R(H,y_R) - (1-\alpha)\,\epsilon_{UR}(H,y_{UR})\right|$$

$$\leq \alpha\left(|\epsilon_R(H,y_R) - \epsilon_{UR}(H,y_{UR})| + |\epsilon_R(H,\widehat{y}_R) - \epsilon_{UR}(H,\widehat{y}_{UR})|\right)$$

$$+ |\epsilon_{UR}(H,\widehat{y}_{UR}) - \epsilon_{UR}(H,y_{UR})|$$

$$\leq \alpha\left(d_{n\Delta n}(D_R, D_{UR}) + \varepsilon + \widehat{\varepsilon}\right) + \rho_{UR}, \tag{22}$$

where the ideal risk in this hypothesis $H$ is the combinatorial error of the ideal joint hypothesis $\varepsilon = \epsilon_R(H^*, y_R) + \epsilon_{UR}(H^*, y_{UR})$ with $H^* = \arg\min_H\left(\epsilon_R(H, y_R) + \epsilon_{UR}(H, y_{UR})\right)$; $\widehat{\varepsilon} = \epsilon_R(H^*, \widehat{y}_R) + \epsilon_{UR}(H^*, \widehat{y}_{UR})$ is the predicted risk; the distribution discrepancy between reliable and unreliable subdomains is $d_{n\Delta n}(D_R, D_{UR}) = 2\sup_{H,H'\in n}\left|\mathbb{E}_{x_R\sim D_R}\left[H(x_R)\neq H'(x_R)\right] - \mathbb{E}_{x_{UR}\sim D_{UR}}\left[H(x_{UR})\neq H'(x_{UR})\right]\right|$; and $\rho_{UR}$ is the predicted label rate of $\widehat{y}_{UR}$, $\rho_{UR} = \epsilon_{UR}(\widehat{y}_{UR}, y_{UR})$. Then, we set $\epsilon_1 = |\epsilon_R(H, y_R) - \epsilon_{UR}(H, y_{UR})|$, $\epsilon_2 = |\epsilon_R(H, \widehat{y}_R) - \epsilon_{UR}(H, \widehat{y}_{UR})|$, and $\epsilon_3 = |\epsilon_{UR}(H, \widehat{y}_{UR}) - \epsilon_{UR}(H, y_{UR})|$, making $\left|\epsilon_{\widehat{T}}(H,\widehat{y}_T) - \epsilon_T(H,y_T)\right| \leq \alpha(\epsilon_1 + \epsilon_2) + \epsilon_3$. By applying the triangle inequality for classification errors [56] as presented in **Lemma 1**, we can prove the upper bound of $\epsilon_1$, $\epsilon_2$, and $\epsilon_3$.

**Lemma 1.** For any hypotheses $H_1$, $H_2$, and $H_3$ in class $n$,
$$\epsilon(H_1, H_2) \leq \epsilon(H_1, H_3) + \epsilon(H_2, H_3). \tag{23}$$

Therefore, for $\epsilon_1$, we can prove that:

$$\epsilon_1 = |\epsilon_R(H, y_R) - \epsilon_{UR}(H, y_{UR})|$$

$$\leq |\epsilon_R(H, y_R) - \epsilon_R(H, H^*)| + |\epsilon_{UR}(H, H^*) - \epsilon_{UR}(H, y_{UR})| + |\epsilon_R(H, H^*) - \epsilon_{UR}(H, H^*)|$$

$$\leq \epsilon_R(H^*, y_{UR}) + \epsilon_{UR}(H^*, y_{UR}) + |\epsilon_R(H, H^*) - \epsilon_{UR}(H, H^*)|$$

$$\leq \frac{1}{2}d_{n\Delta n}(D_R, D_{UR}) + \varepsilon \tag{24}$$

Similarly, for $\epsilon_2$,

$$\epsilon_2 = |\epsilon_R(H, \widehat{y}_R) - \epsilon_{UR}(H, \widehat{y}_{UR})|$$

$$\leq |\epsilon_R(H, \widehat{y}_R) - \epsilon_R(H, H^*)| + |\epsilon_{UR}(H, H^*) - \epsilon_{UR}(H, \widehat{y}_{UR})| + |\epsilon_R(H, H^*) - \epsilon_{UR}(H, H^*)|$$

$$\leq \epsilon_R(H^*, \widehat{y}_R) + \epsilon_{UR}(H^*, \widehat{y}_{UR}) + |\epsilon_R(H, H^*) - \epsilon_{UR}(H, H^*)|$$

$$\leq \frac{1}{2}d_{n\Delta n}(D_R, D_{UR}) + \epsilon_R(H^*, \widehat{y}_R) + \epsilon_{UR}(H^*, \widehat{y}_{UR})$$

$$\leq \frac{1}{2}d_{n\Delta n}(D_R, D_{UR}) + \widehat{\varepsilon}. \tag{25}$$

For $\epsilon_3$,
$$\epsilon_3 = |\epsilon_{UR}(H, \widehat{y}_{UR}) - \epsilon_{UR}(H, y_{UR})| \leq \epsilon_{UR}(\widehat{y}_{UR}, y_{UR}) = \rho_{UR}. \tag{26}$$

By proving $\epsilon_1$, $\epsilon_2$, and $\epsilon_3$, we can derive **Theorem 1**,

$$\left|\epsilon_{\widehat{T}}(H,\widehat{y}_T) - \epsilon_T(H,y_T)\right| \leq \alpha(\epsilon_1 + \epsilon_2) + \epsilon_3$$

$$\leq \alpha\left(\frac{1}{2}d_{n\Delta n}(D_R, D_{UR}) + \varepsilon\right) + \alpha\left(\frac{1}{2}d_{n\Delta n}(D_R, D_{UR}) + \widehat{\varepsilon}\right) + \rho_{UR}$$

$$= \alpha\left(d_{n\Delta n}(D_R, D_{UR}) + \varepsilon + \widehat{\varepsilon}\right) + \rho_{UR}, \tag{27}$$

when the reliable subdomain is mostly correct, $y_R$ and $\widehat{y}_R$ are extremely similar with the ideal risk $\varepsilon$ that is negligibly small [57], in which case $\rho_R \approx 0$, $\widehat{\varepsilon}$ is bounded by the predicted label rate of unreliable subdomain $\rho_{UR}$. Empirical results demonstrate that $d_{n\Delta n}(D_R, D_{UR})$ is usually small across the two subdomains, which plays a significant role in tightening the upper bound though hallucination calibration. Therefore, our method can theoretically reduce the expected error of the model on the target domain.

## C. Algorithm Details

The whole training process is shown in Algorithm 1. In addition, the experimental code and the main code are available in the Supplementary Materials.

**Algorithm 1** CVH-TDN for DABP task.

---

**Input:** Target samples $D_t = \{(x_i)\}_{i=1}^{N_t}$, black-box hard predictions $P_s$, and training model $\mathcal{M}_\theta \in \{F, C\}$;
**Parameter:** The model parameter $\theta$ and the hyperparameters $\lambda$, $\mu$, $\tilde{\mu}_s$, $\phi$, and $r$;
1: **Initialize:** $\mathcal{M}_\theta$ simple tests on $D_t$ to initialize $\theta$ and memory storage $M$; initialize smooth adaptive storage $S$ with $P_s$ and $\tilde{\mu}$ (determined by $\tilde{\mu}_s$ and $\mu$);
2: **while** Adaptation **do**
3:     Get sample batch $B$ using $\mathcal{M}_\theta$ from $D_t$;
4:     **Hallucination Generation:**
5:     Control hallucination image $x_i^g$ generated by $F$ evaluating $x_i$ in $B$ using Eqs. (1)-(4).
6:     **Hallucination Alignment:**
7:     Dynamic divide the samples according to their reliability using Eqs. (6)-(8);
8:     Update $M$ using Eq. (5) to extract the latest information;
9:     Calculate bidirectional alignment weights $w$ using Eq. (9).
10:     **Hallucination Calibration:**
11:     Update $S$ and $\tilde{\mu}$ using Eq. (13) to mimic the brain cognitive processes;
12:     Retrieve $M$ to explore the relationship between features and space and calculate the spatial similarity.
13:     **Model Training:**
14:     Optimize target model $\mathcal{M}_\theta$ by minimizing Eq. (18).
15: **end while**

---

Table 6: Accuracies (%) on the *DomainNet* using the ResNet-50 backbone. The rows represent the source domain and the columns represent the adapted target domain.

| **ResNet** | clp | inf | pnt | qdr | rel | skt | Mean | **DINE** | clp | inf | pnt | qdr | rel | skt | Mean |
|---|---|---|---|---|---|---|---|---|---|---|---|---|---|---|---|
| clp | — | 16.5 | 36.0 | 10.1 | 52.8 | 41.8 | 31.4 | clp | — | 12.1 | 29.6 | 11.1 | 60.4 | 37.3 | 29.4 |
| inf | 32.1 | — | 32.0 | 2.7 | 47.4 | 26.4 | 28.1 | inf | 29.5 | — | 37.6 | 3.4 | 53.8 | 26.5 | 30.1 |
| pnt | 29.6 | 23.2 | — | 4.9 | 36.7 | 27.8 | 24.4 | pnt | 37.3 | 12.9 | — | 4.2 | 60.5 | 34.7 | 29.9 |
| qdr | 11.2 | 1.1 | 1.9 | — | 4.3 | 7.7 | 5.3 | qdr | 9.4 | 0.7 | 3 | — | 8.3 | 6.6 | 5.6 |
| rel | 48.2 | 19.6 | 47.9 | 4.3 | — | 35.6 | 31.1 | rel | 45.1 | 14.4 | 49.7 | 5.5 | — | 35.0 | 29.9 |
| skt | 49.1 | 13.5 | 35.5 | 11.5 | 47.1 | — | 31.3 | skt | 43.3 | 10.0 | 39.3 | 11.6 | 57.2 | — | 32.2 |
| Mean | 34.0 | 14.8 | 30.7 | 6.7 | 37.7 | 27.9 | 25.3 | Mean | 32.9 | 10.0 | 31.8 | 7.2 | 48.0 | 28.0 | 26.2 |

| **BETA** | clp | inf | pnt | qdr | rel | skt | Mean | **CVH-TDN** | clp | inf | pnt | qdr | rel | skt | Mean |
|---|---|---|---|---|---|---|---|---|---|---|---|---|---|---|---|
| clp | — | 13.4 | 41.2 | 13.0 | 61.8 | 41.1 | 34.1 | clp | — | 19.0 | 41.8 | 11.3 | 57.2 | 42.7 | 34.4 |
| inf | 34.9 | — | 41.6 | 3.7 | 56.8 | 30.7 | 33.6 | inf | 37.8 | — | 39.9 | 2.9 | 53.5 | 32.1 | 33.2 |
| pnt | 47.3 | 18.4 | — | 3.2 | 62.5 | 41.9 | 34.7 | pnt | 45.2 | 18.9 | — | 4.0 | 60.3 | 37.9 | 33.3 |
| qdr | 11.7 | 0.9 | 2.1 | — | 9.1 | 8.1 | 6.4 | qdr | 15.0 | 0.9 | 2.9 | — | 6.0 | 11.1 | 7.2 |
| rel | 46.5 | 15.8 | 50.9 | 5.6 | — | 37.7 | 31.3 | rel | 52.7 | 21.7 | 52.5 | 5.7 | — | 39.2 | 34.4 |
| skt | 47.3 | 12.3 | 42.3 | 14.8 | 59.9 | — | 35.3 | skt | 54.1 | 17.3 | 44.3 | 12.4 | 54.3 | — | 36.5 |
| Mean | 37.5 | 12.2 | 35.6 | 8.1 | 50.0 | 31.9 | 28.2 | Mean | 41.0 | 15.6 | 36.3 | 7.3 | 46.3 | 32.6 | **29.8** |

# D. Specific Dataset Details

Four standard benchmark datasets are used for evaluating our method and comparison, including *Office-31* [27], *Office-Home* [28], *VisDA-17* [29], and *DomainNet* [30]. *Office-31* is a small-scale benchmark dataset, which contains 4,110 images with 31 categories from 3 domains, Amazon (A), Dslr (D), and Webcam (W). *Office-Home* is a widely used medium-scale benchmark that contains a total of 15.5K images with 65 categories from 4 distinct domains, Real World (R), Clipart (C), Art (A), and Product (P). *VisDA-17* is a challenging large-scale benchmark with 12 categories that include 152k synthetic source domain images and 55k target images of real objects, presenting a greater challenge due to a large synthetic-to-real domain gap. *DomainNet* is the largest domain adaptation benchmark dataset, which consists of about 600K with 345 categories across 6 domains: Clipart (clp), Infograph (inf), Painting (pnt), Quickdraw (qdr), Real (rel), and Sketch (skt).

# E. Additional Dataset *DomainNet*

For the *DomainNet* dataset, we compare our algorithm with previous SOTA methods [11, 14] under the ResNet-50 backbone. As shown in Table 6, compared with other algorithms, CVH-TDN achieves the highest average accuracy of 29.8% on the *DomainNet*, which is full of on-hard tasks (whose source-only accuracies are below 65%). These results demonstrate that, in the on-hard tasks, our

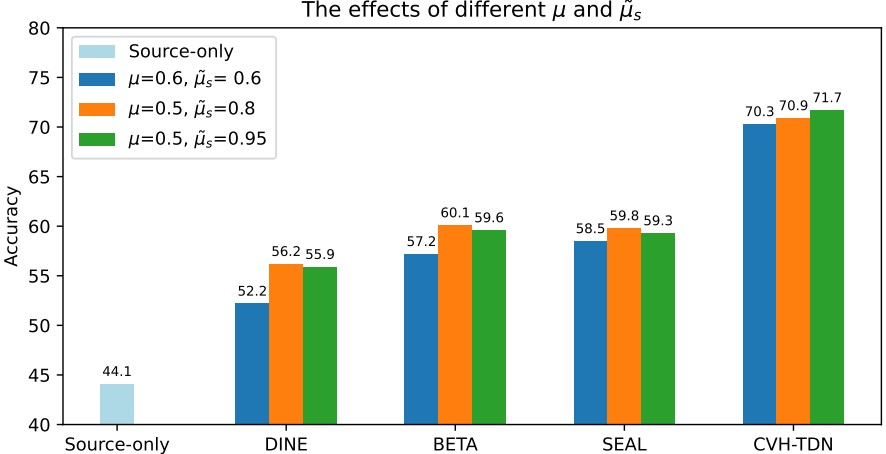

Figure 8: The effect display of different $\mu$ and $\tilde{\mu}_s$ in the A→C subtask of the *Office-Home*. Each result is reported when the best accuracy is achieved.

spatial exploration method based on hallucination control is more effective than methods that rely on suppressing specific sample information.

## F. More Parameter Analysis and Motivation Demonstration

As shown in Figure 8, we report the performance of different methods [11, 14, 31] under various conditions of $\mu$ and $\tilde{\mu}_s$ on the *Office-Home*. In the brain cognitive processes [58], when humans encounter something they have never seen before, their brain cognition is primarily influenced by external information, with a small portion derived from their own understanding. And as their understanding deepens, they can balance the weight of external information and their own understanding, increasingly trusting their own understanding of the matter. In this work, we mimic the brain cognitive process to improve the learning and updating process of Eq. (13), where $\tilde{\mu}_s$ simulates the initial external information and $\mu$ simulates human understanding. Additionally, when $\mu = \tilde{\mu}_s = 0.6$, it indicates the use of the previously common adaptive label smoothing update [11] instead of our update strategy. Experimental results show that this improvement significantly enhances the compared DABP methods.

Moreover, we conducted further analytical experiments to demonstrate the advantage of our method for more effectively leveraging both reliable and unreliable samples compared to existing approaches. First, we elaborate on some concepts: the higher the model prediction accuracy of a class, the higher the proportion of reliable samples in that class. Meanwhile, the samples exhibiting features that clearly distinguish their class tend to have a higher probability of being reliable. Because different methods have different strategies for discriminating reliability, we have selected sample instances with consistent initial reliability judgments between BETA [14] and SEAL [31]. Specifically, in Figure 9, the classes corresponding to high-reliability samples are: Bus, Skateboard, and Person; and the classes corresponding to low-reliability samples are: Car, Motorcycle, and Truck.

As shown in Figure 9, the samples corresponding to the classes of Bus and Skateboard were consistently regarded as reliable samples during the training of BETA and SEAL. They focus on learning from these reliability samples, and Figure 9 demonstrates that they effectively concentrate and lock the feature regions of interest on the salient features. Meanwhile, some low-reliability samples (corresponding to the class Car) have also been correctly rectified from the wrong judgments of the black-box predictors through high sample knowledge. However, for some low-reliability samples (corresponding to the class Motorcycle and Truck), the previous methods all failed: BETA didn't lock onto the effective features, and SEAL mistakenly locked onto the features of other classes. In addition, SEAL even misjudged some samples that were judged as high-reliability samples (corresponding to the class Person) as belonging to other classes. These observations are sufficient to show the limitations of using high-reliability samples to constrain the attention-locked area. Moreover, the

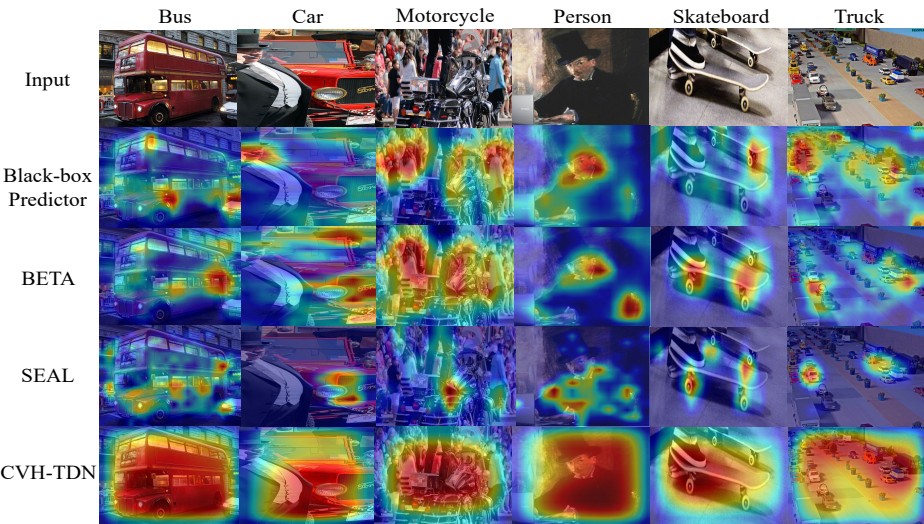

Figure 9: The visualization results of the heat map on the *VisDA-17*. The redder the area, the higher the model's level of attention.

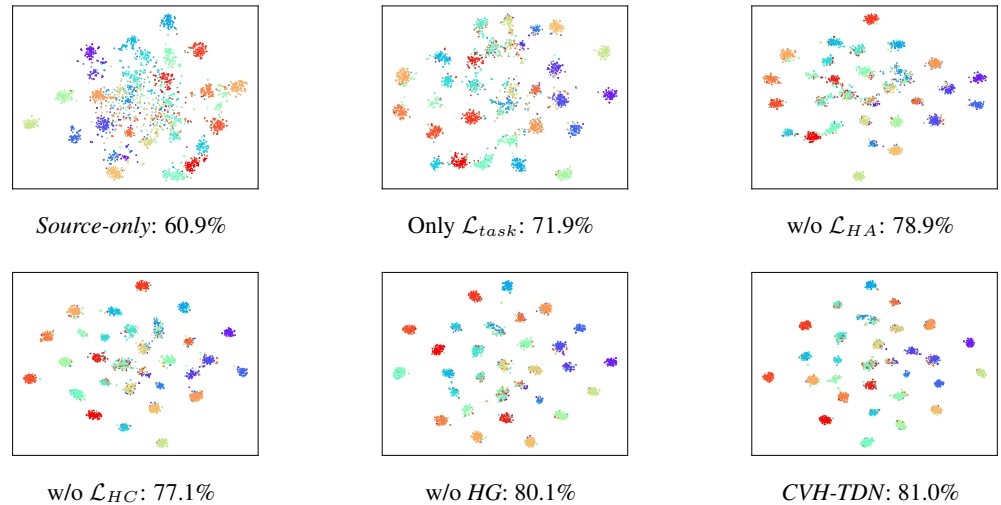

Figure 10: The ablation visualizations of the feature distribution in the W→A subtask of the *Office-31* using t-SNE [12]. Herein, the points represent target samples and the different colors correspond to their true classifications.

feature visualization of Figure 1 can also serve as evidence for the inadequacy of their clustering ability.

In our method, driven by the hallucination alignment, all type-$UR$ samples will become samples of type $R$ in the middle of training. We conceptually blurred the distinction between high-reliability and low-reliability samples during the learning process: samples are no longer constrained by high- or low-reliability ones. As shown in Figures 4 and 9, CVH-TDN enhances the model's reasoning ability by expanding the coverage area of the model's region of interest through hallucination generation and hallucination alignment, resulting in more effective use of all samples.

## G. More Ablation Visualization and Experimental Comparison

In Figure 10, we present the t-SNE visualizations [12] of the ablation study on the *Office-31*. Experimental results show that each component of our method improves the discrimination capacity of classes. It is worth noting that, in the small-scale dataset *Office-31*, the impact of removing HG

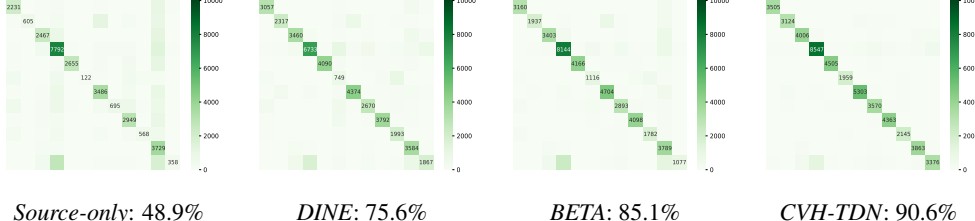

| Source-only: 48.9% | DINE: 75.6% | BETA: 85.1% | CVH-TDN: 90.6% |

Figure 11: Classification visualization with Confusion Matrix to compare different methods on the *VisDA-17*. (Zooming in for a clear view)

Table 7: Results of cost comparison on the *VisDA-17* with the ResNet-101 backbone.

| Method | Time (s/epoch) | Space (MiB) | Accuracy (%) |
|--------|----------------|-------------|--------------|
| DINE | 124s | 9881MiB | 75.6 |
| BETA | 1101s | 20247MiB | 85.1 |
| SEAL | - | Over 24G | 89.2 |
| CVH-TDN | 205s | 10589MiB | 90.6 |

or $\mathcal{L}_{HA}$ is not as significant as shown in the heatmap visualizations [38] of the large-scale dataset *VisDA-17* in Figure 4. This indicates that in datasets with small domain gaps, an efficient clustering algorithm is more important than an algorithm designed to prevent model overfitting, while the opposite holds for datasets with large domain gaps. Furthermore, $\mathcal{L}_{HA}$ focuses on enhancing the model reasoning ability, while $\mathcal{L}_{HC}$ emphasizes improving the class discrimination ability.

For a fair comparison, we set the same running conditions (*e.g.*, batch size = 64, num workers = 4, *etc.*) in the compared works on a machine with an NVIDIA GeForce RTX4090 GPU. Figure 11 shows the Confusion Matrix visualization, in which our method consistently outperforms in discriminating target samples of each class on the *VisDA-17*. This demonstrates that exploring the spatial relationships among samples by controlling hallucinations is more effective in improving class discrimination ability than suppressing specific sample information.

In Table 7, we record the average runtime cost, the maximum GPU space usage, and the best accuracy of each comparison method. When adapting *VisDA-17*, it is worth noting that BETA [14] is divided into two stages that are highly computationally intensive: the first stage is the initialization, which requires initialization of the two models due to their mutually-distilled network structures; the second stage is the two-step process, which requires distillation and fine-tuning for each epoch. SEAL [31] is highly resource-intensive, and its official code cannot complete the adaptation task on *VisDA-17* under the same conditions with 24GB GPU memory. Compared to other methods, CVH-TDN calculates the model area of interest and generates attention-specific masking blocks with minimal cost, which does not require fine-tuning or a computationally expensive generation network. Moreover, our memory structure $M$ uses 57.05 MiB of GPU space usage and our smooth adaptive storage $S$ uses 2.54 MiB of GPU space usage. Since they do not require gradient storage or updates during training, their GPU space usage is nearly negligible.

To ensure robustness, we report the performances across multiple runs with different random seed initializations. As shown in Table 8, we maintained a small average accuracy gap (0.4%) among different seeds, reflecting the stability and superiority of our method.

Table 8: Results under different random seeds on the *Office-Home* with the ResNet-50 backbone.

| Seed | A→C | A→P | A→R | C→A | C→P | C→R | P→A | P→C | P→R | R→A | R→C | R→P | Mean |
|------|-----|-----|-----|-----|-----|-----|-----|-----|-----|-----|-----|-----|------|
| 2022 | 71.7 | 88.4 | 83.6 | 70.1 | 86.7 | 82.8 | 70.8 | 69.2 | 83.3 | 74.7 | 74.2 | 91.6 | 78.9 |
| 2023 | 72.2 | 89.3 | 83.7 | 69.9 | 87.7 | 82.9 | 69.6 | 68.7 | 83.3 | 74.9 | 73.8 | 91.6 | 79.0 |
| 2024 | 71.7 | 88.7 | 83.3 | 69.7 | 86.1 | 83.3 | 70.2 | 68.9 | 83.7 | 73.8 | 72.6 | 91.3 | 78.6 |
| 2025 | 72.1 | 88.6 | 83.5 | 70.0 | 87.9 | 82.4 | 70.1 | 67.2 | 83.7 | 75.4 | 72.1 | 91.4 | 78.7 |

## H. Broader Impacts and Limitations

Our work CVH-TDN focuses on the problem of Domain Adaptation of Black-Box Predictors (DABP), which provides better data privacy protection with more flexible portability compared with other DA settings. Inspired by research in pathology and neuroscience, CVH-TDN is specifically designed for the DABP classification task. While its effectiveness has been demonstrated through extensive experiments and its theoretical soundness established, its applicability to other tasks remains an open question. Therefore, we plan to further explore the practical utility of this algorithm in a broader range of task scenarios.

