# OpenReview forum: "Controlled Visual Hallucination via Thalamus-Driven Decoupling Network for Domain Adaptation of Black-Box Predictors"
_NeurIPS.cc/2025/Conference — NeurIPS 2025 poster_

### Official Review · Reviewer_ATW6 · 2025-06-18

**Clarity:** 2
**Significance:** 2
**Originality:** 3
**Rating:** 4
**Confidence:** 4

**Summary:**

This paper tackles the task of Domain Adaptation of Black-box Predictors (DABP), which is practical in applications.
The authors claim that previous DABP methods face two major challenges: premature overfitting from over-relying on reliable samples, and neglecting valuable information in unreliable ones. To counter this, the authors propose CVH-TDN, which introduces a thalamus-driven decoupling network to visual tasks, leveraging its connection to hallucination for controlled sample generation in feature space. Comprising Hallucination Generation, Alignment, and Calibration, CVH-TDN explores spatial relationships between samples and hallucinations. Experiments on several datasets show improved performance.

**Questions:**

- As the patch size is $p$, is $z_{i,j}^k$ a matrix  or a scalar?
- In Eq. (2), how is the "/" performed, as $Z$ is a feature map?
- Could you explain why the consistency between $x^g$ and $x$ in Eqs.(10-12) beniefit model adaptation?
- What is the black box model in the experiments?

**Ethical Concerns:**

["NO or VERY MINOR ethics concerns only"]

**Final Justification:**

After the rebuttal, the motivations and readability are improved. I believe the paper meets the NeurIPS standard.

**Limitations:**

The authors have discussed the limitations in Appendix.

**Paper Formatting Concerns:**

I did not notice any major formatting issues.

**Quality:**

3

**Strengths And Weaknesses:**

## Strength
- Domain Adaptation of Black-box Predictors (DABP) is practical in real-world applications, as we often can only access the black box model provided in the source domain.
- The concept of controlled visual hallucination and thalamus-driving is new to domain adaptation, and the connection of domain adaptation and neural science is interesting.
- The paper is ok to read with a clear figure to illustrate the idea.
- Experiments show improved results over methods previously compared.

## Weakness
- The motivation for introducing the hallucination and thalamus to the black box DA is not clear. I suggest that the authors include the tight connection between DABP and hallucination/thalamus in the paper (introduction and methods). Specifically, why hallucination/thalamus suite for DABP, and how the challenges of DABP could be tackled by TDN.
- Another major weakness is that too many symbols are introduced, making the paper hard to read. I believe many symbols are not necessary. Meanwhile, there lack motivation for the equations. For example, Eqs. (6-12) are not mainly listed without the reasons for why to define such equations.
- Too many hyperparameters are introduced. How to set these hyperparameters?

---

> ### Author Rebuttal · Authors · 2025-07-30
>
> We sincerely appreciate the reviewer’s constructive suggestions and have answered them as follows with our highest respect.
>
> Q1: The motivation for introducing the hallucination and thalamus to the black box DA is not clear. I suggest that the authors include the tight connection between DABP and hallucination/thalamus in the paper (introduction and methods). Specifically, why hallucination/thalamus suite for DABP, and how the challenges of DABP could be tackled by TDN.
>
> A1: Please refer to lines 66-72, we have illustrated and explained the motivation for introducing the hallucination and thalamus to the black-box DA. Meanwhile, the influence of the TDN framework on the implementation of our method and how to address the challenges of DABP are detailed in lines 74–85 and illustrated in Figure 2. To address the reviewer’s concerns and enhance the clarity of our contributions, we present a clearer and more detailed explanation as: In DABP, the black-box predictor resembles a cognitively impaired “person” who possesses only partial knowledge and is prone to errors. As a result, the target model’s learning process is inevitably influenced by this noisy guidance from the black-box predictor. To address this issue, we draw inspiration from neuroscience, where such conditions are categorized as hallucination disorders: situations in which useful and interfering information are blended, preventing the brain from making reliable decisions. The TDN offers a comprehensive analysis of the full cycle of hallucination disorders and how to address them. Based on this, prior work [1] divides the TDN process into three stages from the perspective of human attention: hallucination manifestations, human responses, and lesion treatment. Correspondingly, we divide CVH-TDN into three core modules that align with the three stages: Hallucination Generation (HG), Hallucination Alignment (HA), and Hallucination Calibration (HC). Guided by the key cognitive impairments, HG generates hallucination images by simulating the locations of hallucination pathology, thereby masking some noise from the black-box predictor and facilitating subsequent HA and HC to enhance the model’s learning ability using these images. HA enhances feature recognition by mimicking how humans manage cognitive impairments, while HC draws on principles of neurological therapy to guide unreliable data through reasoning based on reliable feature representations.
>
>
> Q2: Another major weakness is that too many symbols are introduced, making the paper hard to read. I believe many symbols are not necessary. Meanwhile, there lack motivation for the equations. For example, Eqs. (6)-(12) are not mainly listed without the reasons for why to define such equations.
>
> A2: We thank the reviewer for the insightful feedback. In the revised version, we have simplified symbols to make the paper easy understand as possible. For example, we have removed the sorting operation and modified the original Eq. (9) to $w=\exp (\log (sim({{M}^{(2)}},norm({{Z}^{g}}))))$.
>
> For the motivation of the equations, we have provided detailed explanations for the motivation behind each equation. Eqs. (6)–(8) are designed to clarify the operations in Eq. (5), and their purpose directly supports the design of Eq. (5). The motivation of Eq. (5) is to control the movement direction of the samples on the feature space by acquiring the sample state immediately, which has been elaborated in detail in lines 196-200. Eq. (9) is to assign different weights by pairing features based on the similarity between the original image and its corresponding generated image, which has been explained in lines 211-212. Eq. (10) encourages samples with similar features to be aligned with their generated counterparts in the feature space, as detailed in lines 221-222. Eq. (11) calibrates Eq. (10) and reduces the noise caused by common features to prevent overfitting of the model, as detailed in lines 222-223. Finally, Eq. (12) integrates Eqs. (10) and (11), as detailed in lines 220. Therefore, the motivations behind all proposed equations have been clearly explained in this paper.
>
> Q3: Too many hyperparameters are introduced. How to set these hyperparameters?
>
> A3: Although our method involves several hyperparameters, most of them are fixed, and we have conducted comprehensive analyses for each to ensure their effectiveness. Please refer to Figure 4, we have conducted a detailed parameter analysis on our hyperparameter $\lambda$. Please refer to Figure 5, we have conducted parameter analysis on our hyperparameters $r$ and $\phi$ on the VisDA-17. Therefore, we recommend using $\lambda$=10%, $r$=0.5, $\phi$=6. In all scenarios, we keep these parameter settings the same. Please refer to Figure 8 and lines 692-702, the hyperparameters $\mu $ and $\tilde{\mu}_s$ are task-specific to DABP, yet their transferability and generality have been validated through integration with existing methods, thereby advancing previous learning strategies in DABP. The parameter sensitivity analysis is presented in Figure 8, our method achieves the best overall performance when $\mu = 0.5$ and $\tilde{\mu}_s = 0.95$, while it shows the strongest improvement over other methods when $\mu = 0.5$ and $\tilde{\mu}_s = 0.8$. For other common parameters, refer to the Implementation Details section.
>
> Q4: As the patch size is $p$, is $z_{i,j}^{k}$ a matrix or a scalar?
>
> A4: $z^{k}$ is the $k$-th dimensional feature and $z_{i,j}^{k}$ is the feature scalar corresponding to the i-th row and j-th column.
>
> Q5: In Eq. (2), how is the "/" performed, as $Z$ is a feature map?
>
> A5: In Eq. (2), "/" is a division operation, $Z$ is the $K$-dimensional transitional output from the bottleneck layer. $Z$ stores ${z_{h,w}^{k}}$ as a matrix, where each $z_{h,w}^{k}$ is a scalar at the corresponding position in $Z$. The definitions of $h$, $w$, and $k$ are provided in lines 168-170. To conserve writing space in the main text, Eq. (2) is simplified and presented as:
>
> $Att_{i,j} = max\left( Tanh\left(  \sum\limits_n \sum\limits_k a_k^n z_{i,j}^k /Z \right), 0 \right)$.
>
> However, the simplified formulation is not entirely precise and may result in some confusion. The accurate formulation of Eq. (2) is as follows:
>
> $Att_{i,j} = max\left( Tanh\left(  \sum\limits_n \sum\limits_k a_k^n z_{i,j}^k /\sum\limits_{h}\sum\limits_{w}{\sum\limits_{k}{z_{_{h,w}}^{k}}}) \right), 0 \right)$.
>
> We consider that the simplified formulation of Eq. (2) might have caused confusion for the reviewer. To clarify, we plan to add the detailed explanation of $Z$ and modify the original Eq. (2) to the accurate formulation in the final version.
>
> Q6: Could you explain why the consistency between $x$ and $x^{g}$ in Eqs. (10-12) beniefit model adaptation?
>
> A6: According to previous studies [1, 2], aligning different versions of generated images with the original ones can enhance model generalization. Therefore, how to generate informative images and efficiently align them with the original images has become a highly active research direction in recent years. Our HG is to generate informative images, and our HA is to efficiently align generated images with the original images. We use contrast learning in HA to explore the spatial relationship between the target sample $x$ and the corresponding generated image $x^{g}$. As shown in lines 221-223, in HA, Eq. (10) encourages samples $x$ whose features are similar to be aligned with the generated version $x^{g}$ in the feature space; Eq. (11) calibrates Eq. (10) and reduces the noise caused by common features to prevent overfitting of the model. Eq. (12) integrates Eq. (10) and Eq. (11) and serves as the target optimization loss of HA. As shown in Figure 4 and Table 5, HA can effectively improve model generalization only when combined with HG. As is well known, enhancing the generalization ability of a model can most directly improve its adaptability.
>
> Q7: What is the black box model in the experiments?
>
> A7: In our paper, we do not mention the term "black box model". We assume that the reviewer’s question might be aiming to clarify what is meant by a "black-box predictor" in this context. In DABP setting, it adapts a model using only the unlabeled target data and a black-box predictor trained on the source domain, e.g., an API service in the cloud. Please refer to Figure 6, the black-box predictor refers to a model trained solely on source domain data and deployed on a cloud server, where its internal architecture is inaccessible and predictions can only be obtained via an API interface.
>
> We appreciate the reviewer’s feedback and hope our response has addressed the reviewer’s concerns.
>
> [1] CyCADA: Cycle-consistent adversarial domain adaptation. In ICML, 2018.
>
> [2] Vdm-da: Virtual domain modeling for source data-free domain adaptation. TCSVT, 2022.

---

> > ### Comment · Reviewer_ATW6 · 2025-08-02
> >
> > I thank the authors for the response. Regarding Q7, I mean that the architecture of the black box predictor should be stated, which I possibly missed. That being said, I am happy to increase the score to 4.

---

> ### Author Response · Authors · 2025-08-03
> **Thank you for your active involvement and prompt feedback**
>
> Thank you for your active involvement and prompt feedback during the discussion phase. We are pleased to know that our work has received your acceptance. For the architecture of the black-box predictor, please refer to lines 282-284. We follow the specification for training the black-box predictor in the source domain as specified in [1] to ensure fair comparisons, where ResNet-50 is used for Office-31, Office-Home, and DomainNet, while ResNet-101 is used for VisDA-17. If there are any remaining concerns, we would be pleased to provide further clarification. Your constructive review has undoubtedly enhanced the quality of our paper, and we sincerely appreciate your contribution.
>
> [1] DINE: Domain Adaptation from Single and Multiple Black-box Predictors. In CVPR, 2022.

---

### Official Review · Reviewer_SF5Z · 2025-06-29

**Clarity:** 3
**Significance:** 3
**Originality:** 3
**Rating:** 4
**Confidence:** 2

**Summary:**

The paper presents a novel approach for Domain Adaptation of Black-box Predictors (DABP) through a framework called Controlled Visual Hallucination via Thalamus-Driven Decoupling Network (CVH-TDN). CVH-TDN introduces three main components: Hallucination Generation, Hallucination Alignment, and Hallucination Calibration, designed to enhance the model's ability to generalize by effectively utilizing both reliable and unreliable samples. Extensive experiments demonstrate that CVH-TDN achieves state-of-the-art performance across multiple benchmarks.

**Questions:**

- How does the thalamus-inspired design specifically address the limitations of prior DABP methods? Could the biological analogy be simplified for broader ML audiences?
- How sensitive is the performance of CVH-TDN to the choice of hyperparameters? Are there recommended settings based on the experiments?

**Ethical Concerns:**

["NO or VERY MINOR ethics concerns only"]

**Final Justification:**

Most of my concerns have been addressed, so I am willing to maintain my original positive score.

**Limitations:**

yes

**Quality:**

3

**Strengths And Weaknesses:**

Strengths:
- The introduction of a thalamus-driven decoupling network in the context of DABP is innovative and offers a unique perspective on handling domain adaptation challenges.
- The three-module design (HG, HA, HC) is well-structured and addresses key limitations of prior work (e.g., overfitting on reliable samples and ignoring unreliable ones).
- The authors conduct extensive experiments on four standard benchmarks, providing strong evidence of the effectiveness of their proposed method.

Weaknesses:
- The proposed method may be complex, which could hinder its adoption in practical applications. Simplifying the framework or providing clearer guidelines for implementation might help.
- While the paper discusses computational efficiency, it would benefit from more detailed comparisons of runtime and resource usage against other methods.

---

> ### Author Rebuttal · Authors · 2025-07-30
>
> We sincerely appreciate the reviewer’s constructive suggestions and believe that the additional analysis and explanations significantly improve the quality of our submission.
>
> Q1: The proposed method may be complex, which could hinder its adoption in practical applications. Simplifying the framework or providing clearer guidelines for implementation might help.
>
> A1: Although our method may appear complex in design, each component has been streamlined into a plug-and-play module for practical implementation. Specifically, HG is an integrated generation module that produces pseudo-images directly from raw inputs; HA is a data integration module that can be combined with HG and takes raw images and a target model as input to compute the optimized loss; HC is a DABP task design module that works in conjunction with HG and HA to accomplish the final objective. In our released code, all three modules are modularized and independently packaged to simplify real-world deployment. In addition, we have revised and simplified Figure 3 to make the method more intuitive. For example, by adjusting the layout of the prediction module and adding more detailed descriptions for each component. Upon acceptance of the paper, we will release the code with clear implementation guidelines to facilitate reproducibility and further research by the community.
>
> Q2: While the paper discusses computational efficiency, it would benefit from more detailed comparisons of runtime and resource usage against other methods.
>
> A2: Please refer to Table 7, we have recorded the average runtime cost, the maximum GPU space usage, and the best accuracy of each comparison method on the VisDA-17 dataset. As described in lines 747-754, when adapting VisDA-17, it is worth noting that BETA is divided into two stages that are highly computationally intensive: the first stage is the initialization, which requires initialization of the two models due to their mutually-distilled network structures; the second stage is the two-step process, which requires distillation and fine-tuning for each epoch. SEAL is highly resource-intensive, and its official code cannot complete the adaptation task on VisDA-17 under the same conditions with 24GB GPU memory. The Results of cost comparison on the VisDA-17 with the ResNet-101 backbone as following:
>
> | Method | Time(s/epoch) | Space(MiB) | Accuracy(%) |
> | --- | --- | --- | --- |
> | DINE | 124s | 9881MiB | 75.6 |
> | BETA | 1101s | 20247MiB | 85.1 |
> | SEAL | - | Over 24G | 89.2 |
> | CVH-TDN | 205s | 10589MiB | 90.6 |
>
> Compared to other methods, CVH-TDN calculates the model area of interest and generates attention-specific masking blocks with minimal cost, which does not require fine-tuning or a computationally expensive generation network. To further assess the resource usage of our memory module, we conducted additional experiments for verification: $M$ uses 57.05 MiB of GPU space usage; $S$ uses 2.54 MiB of GPU space usage. The results demonstrate that our CVH-TDN improves performance through an optimized framework, while also reducing training runtime cost and GPU space usage, highlighting the overall effectiveness and efficiency of our approach.
>
> Q3: How does the thalamus-inspired design specifically address the limitations of prior DABP methods? Could the biological analogy be simplified for broader ML audiences?
>
> A3: Our thalamus-inspired design addresses the limitations of previous DABP methods from two perspectives: the methodological framework and the task-specific level. For the methodological framework, CVH-TDN builds upon a thalamus-driven decoupled network that models the cognition–hallucination relationship, further advancing the exploration of interactions between real and hallucinated samples in the feature space. Different from the previous DABP methods, CVH-TDN leverages sample spatial similarity instead of suppressing targeted sample information, thereby improving model. Please refer to Figure 7, CVH-TDN improves the learning principle of the previous DABP method. For the task-specific level, as detail in lines 230-238, we found that currently commonly used DABP task-specific operations have some limitations and proposed new operations to replace them. As shown in 692-702 and Figure 8, experimental results show that this improvement significantly enhances the compared DABP methods.
>
> To convey the simplified biological analogy to a broader ML audience, we have illustrated and explained “Conceptual figure of CVH-TDN. The black-box predictors resemble agents with prior knowledge but lack the ability to perform targeted discrimination. HG controls mask formation by modeling the location where hallucinations are pathologically generated, driven by the key cognitive impairments observed in TDN. HA improves feature discrimination by simulating how humans deal with cognitive impairments. HC draws on neurotherapeutic principles to guide unreliable data through reasoning using reliable feature representations” in Figure 2. As shown in lines 73-86, We provide a more targeted and accessible explanation tailored to the broader ML audience.
>
> Q4: How sensitive is the performance of CVH-TDN to the choice of hyperparameters? Are there recommended settings based on the experiments?
>
> A4: Please refer to Figure 4, we have conducted a detailed parameter analysis on our hyperparameter $\lambda$. For the hyperparameters $\mu $ and $\tilde{\mu}_s$ of our task-specific improvements, our parameter sensitivity analysis is presented in Figure 8, our method achieves the best overall performance when $\mu = 0.5$ and $\tilde{\mu}_s = 0.95$, while it shows the strongest improvement over other methods when $\mu = 0.5$ and $\tilde{\mu}_s = 0.8$. Please refer to Figure 5, we have conducted parameter analysis on our hyperparameters $r$ and $\phi$ on the VisDA-17. In addition, we have supplemented a more detailed parameter sensitivity analysis as follows:
>
> | CVH-TDN | $r$=0 | $r$=0.3 | $r$=0.5 | $r$=0.8 |
> | --- | --- | --- | --- | --- |
> | $\phi$=3 | 81.21 | 88.31 | 88.23 | 86.42 |
> | $\phi$=4 | 81.42 | 89.12 | 89.50 | 87.56 |
> | $\phi$=5 | 82.11 | 89.99 | 90.07 | 89.62 |
> | $\phi$=6 | 82.46 | 90.22 | 90.58 | 89.53 |
> | $\phi$=7 | 82.51 | 90.51 | 90.46 | 88.52 |
> | $\phi$=8 | 81.99 | 90.36 | 90.40 | 87.39 |
>
> Therefore, we recommend using $\lambda$=10%, $r$=0.5, and $\phi$=6. In all scenarios, we keep these parameter settings the same. For other common parameters, refer to the Implementation Details section.
>
> We appreciate the reviewer’s feedback and hope our response has addressed the reviewer’s concerns.

---

> > ### Comment · Reviewer_SF5Z · 2025-08-02
> >
> > Thank you for your response. Most of my concerns have been addressed, so I am willing to maintain my original positive score.

---

> > > ### Author Response · Authors · 2025-08-03
> > > **Thank you for your active involvement and prompt feedback**
> > >
> > > Thank you for your active involvement and prompt feedback during the discussion phase. We are pleased to know that our rebuttal has addressed most of your concerns. If there are any remaining concerns, we would be pleased to provide further clarification. Your constructive review has undoubtedly enhanced the quality of our paper, and we sincerely appreciate your contribution.

---

### Official Review · Reviewer_7sjS · 2025-07-02

**Clarity:** 3
**Significance:** 3
**Originality:** 3
**Rating:** 5
**Confidence:** 4

**Summary:**

The paper identifies two limitations of the existing black-box source-free UDA methods: (1) they focus too much on obtaining highly reliable samples or estimates, which can lead to overfitting, (2) they typically ignore insightful information in samples that are not reliable, which affects the model's generalizability to the target data. To address these limitations, the paper proposed visual hallucination learning, called controlled visual hallucination via thalamus-driven decoupling network (CVH-TDN), to explore spatial relations between real samples and hallucinations. The CVH-TDN comprises a hallucination generator (for generating hallucination images), hallucination alignment (for aligning target samples and the corresponding hallucination-generated image), and hallucination calibration (for dynamically updating adaptive label smoothing). Four benchmark datasets (Office-31, Office-Home, VisDA-17, and DomainNet) are utilised to conduct experiments, demonstrating the effectiveness of CVH-TDN.

**Questions:**

What is the computational cost of CVH-TDN? Particularly relating to the two memory storage containers adopted by CVH-TDN.

**Ethical Concerns:**

["NO or VERY MINOR ethics concerns only"]

**Final Justification:**

I have read the authors' responses to the issues I raised and the ones raised by the other reviewers. The authors have done well to  address all concerns, as the other reviewers gave positive comments on the authors' responses.  Thus, I maintained my positive score of 'Accept'. Thank you

**Limitations:**

The authors should address the computational cost of CVH-TDN as a limitation, as researchers with limited computational resources may not be able to adopt CVH-TDN for their work.

**Quality:**

3

**Strengths And Weaknesses:**

****Strengths****

(1)  Introducing the thalamus-driven decoupling network (TDN) into black-box source-free UDA is interesting. This is the first time TDN has been applied in a vision task, particularly in source-free domain adaptation, leveraging TDN's connection with hallucination to guide sample generation within a feature space.

(2)  The paper is well-written, well-presented, easy to follow and comprehend. The proposed method, CVH-TDN, is detailed and well-explained.

(3)  The authors conducted extensive experiments on four benchmark datasets to demonstrate the effectiveness of the proposed CVH-TDN. The results show that CVH-TDN achieves an impressive performance across all the datasets and outperforms several SoTA methods in both overall-average accuracy and on-hard task average accuracy. Ablation studies on two benchmark datasets further demonstrate the effectiveness of each component of CVH-TDN.

****Weaknesses****

In addition to hallucination image generation,  CVH-TDN has two memory storages: memory structure ***M*** and smooth adaptive storage ***S***, which indicates that CVH-TDN is computationally expensive. The authors did not address the computational cost of CVH-TDN.

---

> ### Author Rebuttal · Authors · 2025-07-30
>
> We sincerely appreciate the reviewer’s constructive suggestions and have answered them as follows with our highest respect.
>
> Q1: The authors should address the computational cost of CVH-TDN as a limitation, as researchers with limited computational resources may not be able to adopt CVH-TDN for their work.
>
> A1: Please refer to Table 7 in Appendix, we have recorded the average runtime cost, the maximum GPU space usage, and the best accuracy of each comparison method on the VisDA-17 dataset. The Results of cost comparison on the VisDA-17 with the ResNet-101 backbone as following:
>
> | Method | Time(s/epoch) | Space(MiB) | Accuracy(%) |
> | --- | --- | --- | --- |
> | DINE | 124s | 9881MiB | 75.6 |
> | BETA | 1101s | 20247MiB | 85.1 |
> | SEAL | - | Over 24G | 89.2 |
> | CVH-TDN | 205s | 10589MiB | 90.6 |
>
> For a fair comparison, we set the same running conditions (e.g., batch size = 64, num workers = 4, etc.) in the compared works on a machine with an NVIDIA GeForce RTX4090 GPU. Compared with other methods, CVH-TDN improves performance through the optimized framework while reducing training runtime cost and GPU space usage. This enables researchers with limited computing resources to adopt CVH-TDN for their work.
>
> Q2: What is the computational cost of CVH-TDN? Particularly relating to the two memory storage containers adopted by CVH-TDN.
>
> A2: The computational cost of CVH-TDN refers to Q1. For our memory structure $M$ and smooth adaptive storage $S$, we supplemented additional experiments on the VisDA-17 using an NVIDIA GeForce RTX4090 GPU to verify their space costs: $M$ uses 57.05 MiB of GPU space usage; $S$ uses 2.54 MiB of GPU space usage. Since they do not require gradient storage or updates during training, their GPU space usage is nearly negligible.
>
> We appreciate the reviewer’s feedback and hope our response has addressed the reviewer’s concerns.

---

> > ### Comment · Reviewer_7sjS · 2025-08-03
> >
> > I thank the authors for their response. My concerns have been addressed, and I am maintaining my original score.

---

> > > ### Author Response · Authors · 2025-08-03
> > > **Thank you for your active involvement and prompt feedback**
> > >
> > > Thank you for your active involvement and prompt feedback during the discussion phase. We are pleased to know that our rebuttal has addressed your concerns. Your constructive review has undoubtedly enhanced the quality of our paper, and we sincerely appreciate your contribution.

---

### Official Review · Reviewer_juSi · 2025-07-02

**Clarity:** 2
**Significance:** 3
**Originality:** 3
**Rating:** 5
**Confidence:** 3

**Summary:**

The paper aims to propose a better method that mitigates overfitting to reliable samples by generating hallucinated images and enhancing their similarity to the original images when facing the DABP problem.

However, the writing of this paper is not very strong, as it includes many unnecessary descriptions, such as the TDN, which seems to serve more as a narrative device rather than a methodological component. As a result, many key ideas are not explained clearly and concisely.

For example, among the three losses, the first two appear to incorporate adversarial concepts—striving for similarity while simultaneously preventing excessive similarity. Additionally, it is unclear how the unreliable information is actually utilized. From Equation 14, it seems that the method still primarily relies on reliable information. There are many other issues—please refer to the following comments.

**Questions:**

1. In Equation (9), what is the rationale for sorting the weights w? Why is this operation necessary in the context of the method?

2. The loss formulation in Equation (12) appears adversarial in nature. The objectives of the forward and backward losses seem inconsistent. How is balance maintained during optimization, and how does the model effectively learn under such a setting?

3. The calibration section is somewhat unclear. Equation (16) also seems to follow an adversarial formulation. Could the authors clarify the intended mechanism and its theoretical or empirical justification?

4. Why are the losses from the three stages directly combined and used to jointly train a single model for UDA? Would it not be more natural to train the model in separate stages?

5. I am also curious about the API used in this work. If the model is simply distilling from a powerful closed-source model, the performance gains may be primarily attributed to the strength of the source model rather than the proposed method itself.

**Ethical Concerns:**

["NO or VERY MINOR ethics concerns only"]

**Final Justification:**

According to the rebuttal, my concerns and other reviewers' concerns have been answered.

**Limitations:**

Please see the questions.

**Paper Formatting Concerns:**

Good.

**Quality:**

3

**Strengths And Weaknesses:**

Strengths:

1. The introduction of the thalamus-driven concept is interesting and adds a biologically inspired perspective.

2. The proposed method demonstrates a reasonable degree of novelty.

3. The final results are relatively strong and show promising performance.

Weaknesses:

1. The connection between the thalamus-driven concept and the proposed method feels somewhat tenuous and would benefit from a clearer and more refined explanation.

2. The methodology section is difficult to follow, and the current model diagram is not very intuitive. A more focused or simplified illustration highlighting the core idea would help improve clarity.

---

> ### Author Rebuttal · Authors · 2025-07-30
>
> We sincerely appreciate the reviewer’s constructive suggestions and have answered them as follows with our highest respect.
>
> Q1: The connection between the thalamus-driven concept and the proposed method feels somewhat tenuous and would benefit from a clearer and more refined explanation.
>
> A1: To better understand the connection between the Thalamus-driven Decoupling Network (TDN) and CVH-TDN, we present a clearer and more detailed explanation as: In DABP, the black-box predictor resembles a cognitively impaired “person” who possesses only partial knowledge and is prone to errors. As a result, the target model’s learning process is inevitably influenced by this noisy guidance from the black-box predictor. To address this issue, we draw inspiration from neuroscience, where such conditions are categorized as hallucination disorders: situations in which useful and interfering information are blended, preventing the brain from making reliable decisions. The TDN offers a comprehensive analysis of the full cycle of hallucination disorders and how to address them. Based on this, prior work [1] divides the TDN process into three stages from the perspective of human attention: hallucination manifestations, human responses, and lesion treatment. Correspondingly, we divide CVH-TDN into three core modules that align with the three stages: Hallucination Generation (HG), Hallucination Alignment (HA), and Hallucination Calibration (HC). Guided by the key cognitive impairments, HG generates hallucination images by simulating the locations of hallucination pathology, thereby masking some noise from the black-box predictor and facilitating subsequent HA and HC to enhance the model’s learning ability using these images. HA enhances feature recognition by mimicking how humans manage cognitive impairments, while HC draws on principles of neurological therapy to guide unreliable data through reasoning based on reliable feature representations.
>
> Q2: The methodology section is difficult to follow ... highlighting the core idea would help improve clarity.
>
> A2: We appreciate the reviewer’s valuable suggestion. We have revised and simplified the existing model diagram to make it more intuitive. For example, by adjusting the overall proportion of the prediction module and adding more detailed descriptions for each component. We have also revised some representations to make it easy to follow. All the revisions will be included in the final version.
>
> Q3: In Equation (9), what is the rationale for sorting the weights w? Why is this operation necessary in the context of the method?
>
> A3: We thank the reviewer for the insightful feedback. in Eq. (9), the sorting operation is originally introduced to prevent potential information misalignment caused by changes in sample correspondence during training. However, since Eq. (5) inherently ensures one-to-one alignment, further experimental results show that the presence or absence of the sorting step has negligible impact on the final performance. Based on this observation, we have simplified Eq. (9) to a more concise form, denoted as $w=\exp (\log (sim({{M}^{(2)}},norm({{Z}^{g}}))))$.
>
> Q4: The loss formulation in Equation (12) appears adversarial ... how does the model effectively learn under such a setting?
>
> A4: Eq. (12) is composed of bidirectional losses: $L_{HA}^{forward}$ encourages samples whose features are similar to be aligned with the generated version in the feature space; $L_{HA}^{back}$ calibrates $L_{HA}^{forward}$ and reduces the noise caused by common features to prevent overfitting of the model. During the optimization process, the weight $w$ dynamically evaluates the importance of features to determine the influence of $L_{HA}^{forward}$, thereby maintaining a balance between $L_{HA}^{forward}$ and $L_{HA}^{back}$. These descriptions can be found on lines 211-223. To better understand their relationship and support effective model learning in this setting, we have added ablation experiments on Office-31 to validate their correlation:
>
> | Office-31 | A→D | A→W | D→A | D→W | W→A | W→D | Mean |
> | ---- | ---- | ---- | ---- | ---- | ---- | ---- | ---- |
> | CVH-TDN w/o Eq. (12) | 92.9 | 90.2 | 73.9 | 98.6 | 78.9 | 99.4 | 89.0 |
> | CVH-TDN w/o $L_{HA}^{forward}$ | 94.1 | 90.9 | 74.1 | 98.9 | 78.6 | 99.4 | 89.3 |
> | CVH-TDN w/o $L_{HA}^{back}$ | 92.9 | 89.6 | 72.1 | 98.5 | 76.4 | 99.6 | 88.1 |
> | CVH-TDN | 96.4 | 92.8 | 75.6 | 98.9 | 81.0 | 99.6 | 90.7 |
>
> We observe that when only $L_{HA}^{forward}$ is applied, the lack of constraint from $L_{HA}^{back}$ leads to rapid alignment between original and generated samples. This causes the model to be heavily influenced by common features during training, resulting in a potential performance drop. Since $L_{HA}^{back}$ is specifically designed to complement $L_{HA}^{forward}$ by mitigating such interference during alignment, using $L_{HA}^{back}$ alone does not effectively improve the training process, and thus brings little to no performance gain. Therefore, the joint effect of bidirectional losses is essential for enabling efficient learning in this setting.
>
> Q5: The calibration section is somewhat unclear ... its theoretical or empirical justification?
>
> A5: In the calibration section, Eq. (16) can be divided into two parts: Hsc and Cfe. The Hsc is to calibrate samples with similar signature features, while the Cfe is to separate ambiguous samples with similar common features. In Hsc, the high-reliability samples are distinguished by the previous hierarchical strategy to cluster the low-reliability samples that are close in spatial distance. In Cfe, a regularization loss is introduced to enhance the difference in the representation of different samples in the feature space, and the generated hallucination image is forcibly aligned with the original image to guide the separation of different classes of samples. More detailed descriptions of the intended mechanism can be found on lines 239-257. To verify these judgments, as shown in Figure 4, both Hsc and Cfe of $L_{HC}$ contribute to improving the reasoning ability of the model by highlighting the areas that the model focuses on for a given category. When Hsc is removed, the values of the regions of interest in the model are averaged, resulting in a lot of non-main features being captured. When Cfe is removed, the ability to grasp key features is weakened, resulting in some key features not being captured.
>
> Q6: Why are the losses from the three stages ... train the model in separate stages?
>
> A6: Please refer to Figure 6. Unlike UDA, under the DABP setting, we do not have access to the source model and can only obtain noisy target labels via API, which prevents us from performing training based on the source model. From a setup perspective, the DABP training consists of only one stage—adapting the model to the target domain. From a methodological standpoint, CVH-TDN can be divided into two sub-stages: HG corresponds to the hallucination manifestations stage in TDN, while the combination of HA and HG corresponds to the hallucination learning stage. The combined update of loss reflects the practical implementation of our optimization strategy. From both a causal and pathological perspective, the hallucination learning stage necessarily follows the hallucination manifestations stage, during which human responses and lesion treatment occur concurrently. This process closely mirrors how hallucinations are understood and handled in TDN. Training the model in separate stages incurs higher computational costs, and its effectiveness still requires further validation. We plan to explore this direction in future work.
>
> Q7: I am also curious about the API used ... rather than the proposed method itself.
>
> A7: Please refer to Tables 1, 2, 3 and 6, under fair comparison conditions and with the same closed-source model, all results demonstrate that CVH-TDN significantly outperforms previous SOTA methods, highlighting the superiority of our approach. To further investigate whether the performance gain is primarily due to the strength of the source model, we conducted additional experiments under a more powerful multi-source setting and further enhanced the source model using the domain generalization (DG). To ensure a fair comparison, we select the same ResNet-50 for the black-box predictor learning on Office-Home under multi-source DABP setting. We select the DG method Swad [2] mentioned to train the source model as a strong cloud API, while following the DINE [3] and SEAL [4] strategy to simply train a source model as a weak cloud API:
>
> | Office-Home | → Art | → Clipart | → Product | → Real_World | Mean |
> | --- | --- | --- | --- | --- | --- |
> | weak | 54.9 | 49.9 | 69.6 | 76.7 | 62.8 |
> | DINE (weak) | 74.9 | 62.6 | 84.6 | 84.7 | 76.7 |
> | SEAL (weak) | 75.3 | 66.4 | 86.9 | 85.2 | 78.5 |
> | CVH-TDN (weak) | 74.6 | 73.1 | 91.3 | 85.1 | 81.0 |
> | strong | 65.3 | 57.4 | 78.8 | 81.0 | 70.6 |
> | DINE (strong) | 75.4 | 64.2 | 85.1 | 84.9 | 77.4 |
> | SEAL (strong) | 75.7 | 67.1 | 86.8 | 86.1 | 78.9 |
> | CVH-TDN (strong) | 75.0 | 75.3 | 91.8 | 86.7 | 82.2 |
>
> As shown in the above table and the main results throughout the paper, CVH-TDN consistently demonstrates superior performance across different models and settings. While the performance gains are partially influenced by the strength of the source model, the primary contributing factor remains the learning capability of CVH-TDN itself.
>
> We appreciate the reviewer’s feedback and hope our response has addressed the reviewer’s concerns.
>
> [1] Understanding visual hallucinations: A new synthesis. Neur. Bio. Rev, 2023
>
> [2] Swad: Domain generalization by seeking flat minima. In NeurIPS, 2021
>
> [3] DINE: Domain Adaptation from Single and Multiple Black-box Predictors. In CVPR, 2022
>
> [4] A Separation and Alignment Framework for Black-Box Domain Adaptation. In AAAI, 2024

---

> > ### Comment · Reviewer_juSi · 2025-08-03
> > **Reply**
> >
> > I have read all the comments, including those from other reviewers. My concerns have been well solved, so I would like to increase the score.

---

> > > ### Author Response · Authors · 2025-08-04
> > > **Thank you for your active involvement and prompt feedback.**
> > >
> > > Thank you for your active involvement and prompt feedback during the discussion phase. We are pleased to know that our rebuttal has well addressed your concerns. Your constructive review has undoubtedly enhanced the quality of our paper, and we sincerely appreciate your contribution.

---

### Decision · Program_Chairs · 2025-09-17

**Decision:**

Accept (poster)

**Comment:**

This work considers domain adaptation with Black-box Predictors (DABP).  The paper points out that existing black-box source-free UDA methods risk overfitting by relying only on highly reliable samples and neglecting less reliable ones, which harms generalization. To overcome this, it proposes Controlled Visual Hallucination via Thalamus-Driven Decoupling Network (CVH-TDN) to exploit spatial relations between real and hallucinated samples.

Almost all reviewers believe this work is novel and performance is stronge, and can be accepted.

I recommend to accept this work as a poster.